# Espin enhances confined cell migration by promoting filopodia formation and contributes to cancer metastasis

Yan Wang[1,2,7], Peng Shi [ID][3,7 ✉], Geyao Liu[1,2], Wei Chen[4], Ya-Jun Wang[5], Yiping Hu[1,2], Ao Yang[6], Tonghua Wei[1,2], Yu-Chen Chen[5], Ling Liang [ID][6], Zheng Liu [ID][4 ✉], Yan-Jun Liu [ID][5 ✉] & Congying Wu [ID][1,2 ✉]

## Abstract

Genes regulating the finger-like cellular protrusions—filopodia have long been implicated in cancer metastasis. However, depleting the flat lamellipodia but retaining filopodia drastically hampers cell migration on spread surface, obscuring the role of filopodia in cell motility. It has been noticed recently that cells under confinement may employ distinct migratory machineries. However, the regulating factors have mainly been focused on cell blebbing, nuclear deformation and cell rear contractility, without much emphasis on cell protrusions and even less on filopodia. Here, by micropore-based screening, we identified espin as an active regulator for confined migration and that its overexpression was associated with metastasis. In comparison to fascin, espin showed stronger actin bundling in vitro and induced shorter and thicker filopodia in cells. Combining the imaging-compatible microchannels and DNA-based tension probes, we uncovered that espin overexpression induced excessive filopodia at the leading edge and along the sides, exerting force for confined migration. Our results demonstrate an important role for filopodia and the regulating protein—espin in confined cell migration and shed new light on cytoskeletal mechanisms underlying metastasis.

**Keywords** Cancer Metastasis; Confined Migration; Espin; Filopodia
**Subject Categories** Cancer; Cell Adhesion, Polarity & Cytoskeleton

## Introduction

During 2D migration, actin filaments at the cell leading edge assemble to promote cell protrusions such as flat lamellipodia and thin filopodia, which generate protrusive force and sense chemical and physical cues in the microenvironment (Albuschies and Vogel, 2013; Chan and Odde, 2008; Jacquemet et al, 2015; Vasioukhin et al, 2000). Abundant filopodia are characteristic of invasive cancer cells and high expression of filopodia regulators such as fascin and myosin-X are associated with cancer progression (Cao et al, 2014; Huang et al, 2015). Contrary to the majority of lamellipodia formed as an extension of the cell body along the bottom matrix, many 2D cultured cells extend filopodia not only along the leading edge but also towards the dorsal side. Lamellipodia deficiency resulting from Arp2/3 depletion largely abolishes cell motility, while the remaining filopodia-driven cell motility appears to be inefficient (Suraneni et al, 2012; Wu et al, 2012). These long-standing observations have posed the question of the exact role of filopodia in cell motility and how that contributes to cancer metastasis.

Metastasis is the leading cause of cancer-related death. It is a complex process during which cancer cells invade from primary tumors, enter and survive in circulation, then exit the bloodstream and form metastatic tumors at distant tissues. To invade surrounding tissues, cancer cells can remodel the microenvironment via secretion of matrix metalloproteinases (MMPs) that degrade the extracellular matrix (ECM) and can also squeeze through physical constrictions much smaller than their diameters without ECM degradation (referred to as confined migration) (Davidson et al, 2014; Liu et al, 2015; Page-McCaw et al, 2007). Cancer cells experience extensive morphological deformation when subjected to various mechanical constraints such as migrating in dense ECM or squeezing through the endothelium (Paul et al, 2017; Reymond et al, 2013; Roberts et al, 2021). It has been reported that confined migration of cancer cells enhances metastatic capacity (Bera et al, 2022; Fanfone et al, 2022; Yang et al, 2023). Deciphering the molecular mechanisms underlying confined migration may help target cancer metastasis.

Mechanistically, cell migration through constricted space requires dynamic nuclear deformation and forward force generation (Hervas-Raluy et al, 2019; Paul et al, 2017). Under conditions of low-adhesive confinement, mesenchymal tumor cells can

[1]Institute of Systems Biomedicine, School of Basic Medical Sciences, Peking University Health Science Center, 100191 Beijing, China. [2]International Cancer Institute, Beijing Key Laboratory of Tumor Systems Biology, Peking University Health Science Center, 100191 Beijing, China. [3]Cancer Institute, Suzhou Medical College, Soochow University, 215000 Suzhou, Jiangsu, China. [4]The Institute for Advanced Studies, TaiKang Center for Life and Medical Sciences, Hubei Key Laboratory of Cell Homeostasis, College of Life Sciences, Wuhan University, 430072 Wuhan, Hubei Province, China. [5]Shanghai Xuhui Central Hospital, Zhongshan-Xuhui Hospital, Shanghai Key Laboratory of Medical Epigenetics, Institutes of Biomedical Sciences, Fudan University, 200032 Shanghai, China. [6]Department of Biophysics, School of Basic Medical Sciences, Peking University Health Science Center, 100191 Beijing, China. [7]These authors contributed equally: Yan Wang, Peng Shi. ✉E-mail: pengshi@suda.edu.cn; zheng.liu@whu.edu.cn; Yanjun_Liu@fudan.edu.cn; congyingwu@hsc.pku.edu.cn

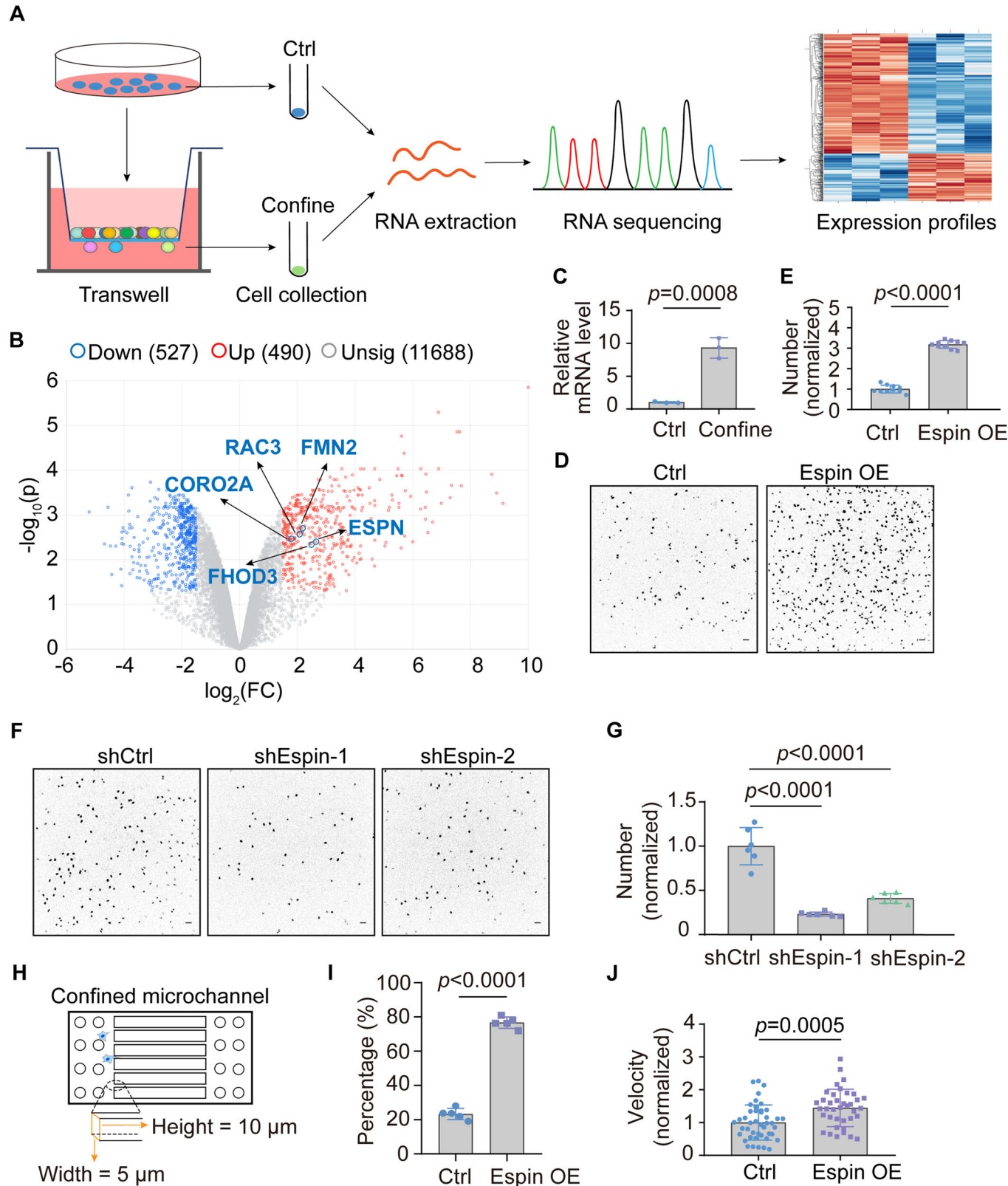

**Figure 1.   Identification of espin as an active regulator for confined migration.**

(A) Diagram showing the process of micropore-based screening. MEF cells were seeded on the top of the transwell with 3 μm pore size and collected at the bottom after 15 h. (B) Interactive volcano plot showing the feature pattern: confine versus ctrl. Several F-actin-associated proteins were circled and annotated. *ESPN* was the corresponding gene name for protein espin. The plot was acquired using NetworkAnalyst. Differential expression was analyzed using DESeq2. log$_2$(FC) > 1.5, P < 0.05. (C) The relative level of *ESPN* mRNA in control and transmigrated cells. Data represent technical replicates and are shown as mean ± SD. $n_{(Ctrl)}$ = 3, $n_{(Confine)}$ = 3. Significance was tested using unpaired Student's *t* test. The experiment was independently replicated once. (D) Representative images of control and espin OE cells on the bottom surface of 5 μm transwell (10 μm long). Scale bar: 50 μm. The experiment was independently replicated three times. (E) Quantification of cell number in (D). Data represent technical replicates and are shown as mean ± SD. $n_{(Ctrl)}$ = 10, $n_{(Espin OE)}$ = 10. Significance was tested using unpaired Student's *t* test. $P_{(Ctrl, Espin OE)}$ = 1e-15. (F) Representative images of bottom shCtrl and espin KD cells on 5 μm transwell. Scale bar: 50 μm. The experiment was independently replicated twice. (G) Quantification of cell number in (F). Data represent technical replicates and are shown as mean ± SD. $n_{(shCtrl)}$ = 6, $n_{(shEspin-1)}$ = 6, $n_{(shEspin-2)}$ = 6. Significance was tested using one-way ANOVA and unpaired Student's *t* test. $P_{(shCtrl, shEspin-1)}$ = 4.54e-6, $P_{(shCtrl, shEspin-2)}$ = 5.7e-5. (H) Schematic representation of the confined microchannel. (I) Quantification of cell percentage of control and espin OE cells entered the microchannels. Data represent biological replicates and are shown as mean ± SD. $N_{(Ctrl)}$ = 5, $N_{(Espin OE)}$ = 5. Significance was tested using paired Student's *t* test. The experiment was independently replicated five times. $P_{(Ctrl, Espin OE)}$ = 5.39e-5. (J) Quantification of nuclear velocity of control and espin OE cells in the microchannels. Data represent technical replicates and are shown as mean ± SD. $n_{(Ctrl)}$ = 44, $n_{(Espin OE)}$ = 38. Significance was tested using unpaired Student's *t* test. The experiment was independently replicated once. Source data are available online for this figure.

undergo a morphological switch to adopt fast ameboid migration driven by membrane blebbing and cell rear contraction (Liu et al, 2015). Perinuclear actin network or reduced lamin A level can induce increased nuclear deformability, consequently facilitating confined migration (Bell et al, 2022; Harada et al, 2014; Thiam et al, 2016). The forward nuclear movement is supported by actomyosin contraction in coordination with microtubule-mediated pulling (Ju et al, 2024; Marks and Petrie, 2022; Renkawitz et al, 2019). Nevertheless, the contribution of cell leading edge and actin-based protrusions is largely under-studied in confined migration.

Espin is characterized as an actin-bundling protein involved in the formation of parallel actin bundles, including stereocilia and filopodia (Bartles et al, 1998; Chou et al, 2011; Loomis et al, 2003). In this study, we have identified espin as an active regulator for confined migration by promoting filopodia formation, along the cell periphery and towards the dorsal surface. In comparison to fascin, espin shows stronger actin-bundling activity in vitro and induces shorter and thicker filopodia in cells. During confined migration, espin overexpressing cells form excessive filopodia at the leading edge and along the sides, exerting force for cells migrating in confined environments. Moreover, espin contributes to cancer metastasis without affecting cell proliferation, invadopodia assembly or cell survival under fluid shear stress. These findings unveil a critical and definitive role of filopodia in cell motility and shed light on the contribution of actin-based protrusions in confined migration and cancer metastasis.

## Results

### Identification of espin as an active regulator for confined migration

To identify potential regulators involved in confined cell migration, we performed transwell migration assay with narrow pore size to simulate native physical confined microenvironment. Transmigrated cells were collected for RNA sequencing (RNA-seq) to analyze transcriptional profile changes in comparison to parental cells (Fig. 1A). In total, 1017 genes were differentially expressed (490 upregulated genes and 527 downregulated genes), among which 27 genes were identified to code actin-associated proteins (Fig. 1B). The ectoplasmic specialization protein (espin), an actin-bundling protein, was retrieved from the upregulation list

(corresponding gene *ESPN*) and verified using real-time quantitative PCR (Fig. 1B,C). As a verification of the RNA-seq data, we found that the number of cells transmigrated through 5 μm pores markedly increased under espin overexpression (OE) while reduced under espin knockdown (KD) compared to control MDA-MB-231 cells (Figs. 1D–G and EV1A,B). To eliminate the possibility that the increased transmigration of espin OE cells was resulted from accelerated cell proliferation, we monitored cell growth rate in parallel and found that altered espin expression had little effect (Fig. EV1C,D). These observations argued that espin promoted confined migration. To better elucidate the underlying mechanism, we fabricated imaging-adaptable microchannels with 5 μm width and 10 μm height to create a confined microenvironment for cells (Fig. 1H) (Wang et al, 2022b). Consistent with the transwell results, more espin OE cells entered the microchannels than control cells, with elevated velocity as shown by the nuclear movement (Figs. 1I,J and EV1E,F), thereby providing additional evidence for the crucial role of espin in promoting confined cell migration.

### Espin expression is positively correlated with cancer malignancy and metastasis

Confined migration has been implicated in tumor metastasis (Bera et al, 2022; Fanfone et al, 2022; Yang et al, 2023). We thus set to investigate whether espin modulated tumor metastasis. We evaluated the expression profiles of espin in tumor and normal samples with the TCGA and GTEx databases (Consortium, 2015) and found that espin was elevated in multiple tumors (Fig. 2A). Moreover, the higher expression level of espin was associated with lower overall survival, especially within the first 150 months, suggesting a correlation between espin and cancer aggressiveness during the early period (Fig. 2B).

We then generated the espin OE B16-F10 cell line and examined the role of espin in tumor growth using a subcutaneous tumor model (Fig. 2C). In accordance with our finding that espin did not promote cell proliferation, we observed that the subcutaneous tumor weight remained similar between control and espin OE groups (Fig. 2D,E). To investigate whether espin regulates cancer metastasis, control and espin OE cells were injected into the tail vein of C57BL/6J mice (Fig. 2F). After 2 weeks following the injection, metastatic tumors on the lung surface were visible as black puncta and counted as metastatic foci (Fig. EV2A). The number of metastatic foci was markedly increased in espin OE group (Fig. 2G,H). These results suggest that

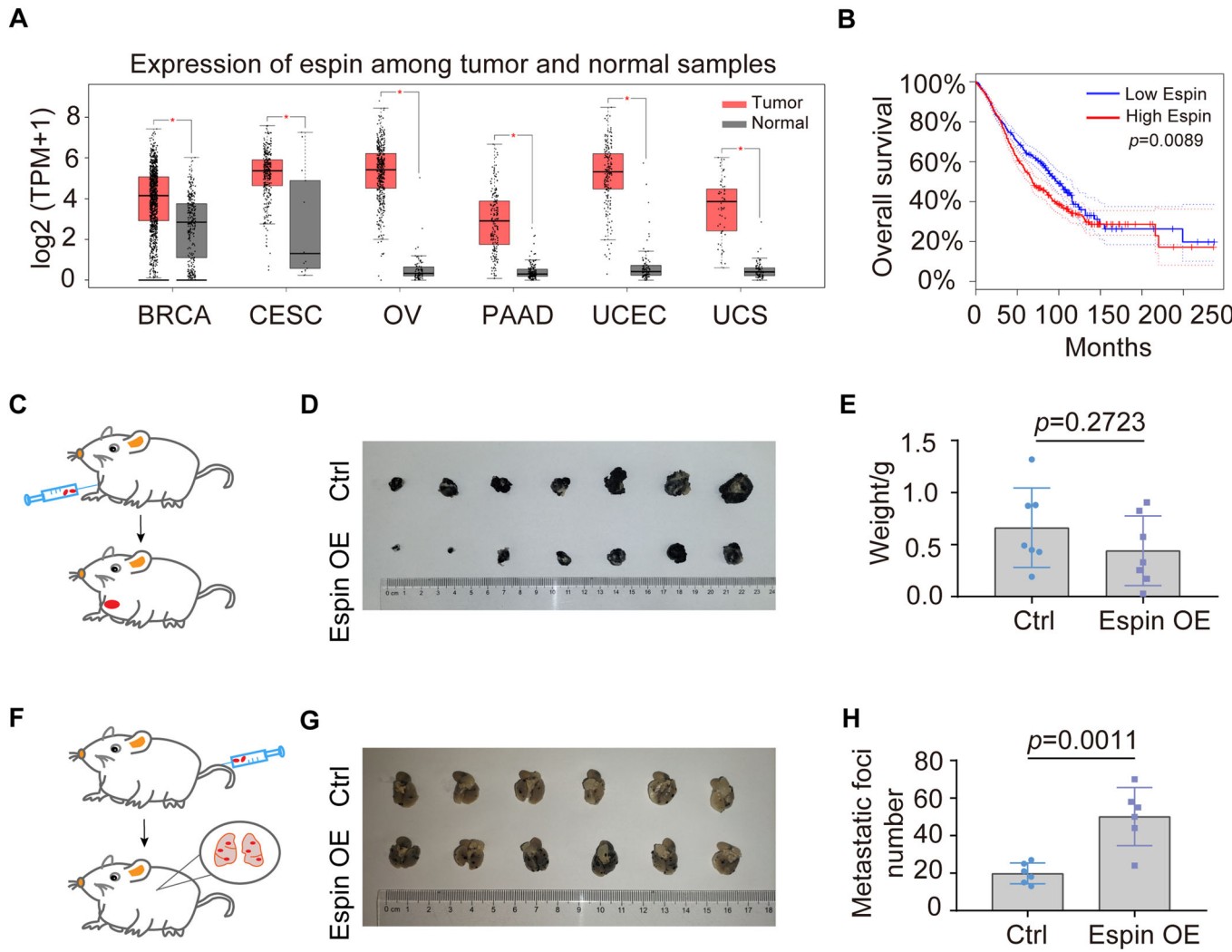

**Figure 2. Espin expression is positively correlated with cancer malignancy and metastasis.**

(A) Expression profiles of espin among TCGA tumor samples (red) and GTEx normal samples (gray) using GEPIA. TPM: transcripts per million, BRCA: breast invasive carcinoma, CESC: cervical squamous cell carcinoma and endocervical adenocarcinoma, OV: ovarian serous cystadenocarcinoma, PAAD: pancreatic adenocarcinoma, UCEC: uterine corpus endometrial carcinoma, UCS: uterine carcinosarcoma. $n_{(BRCA\ Tumor)}$=1085, $n_{(BRCA\ Normal)}$=291, $n_{(CESC\ Tumor)}$=306, $n_{(CESC\ Normal)}$=13, $n_{(OV\ Tumor)}$= 426, $n_{(OV\ Normal)}$=88, $n_{(PAAD\ Tumor)}$=179, $n_{(PAAD\ Normal)}$=171, $n_{(UCEC\ Tumor)}$=174, $n_{(UCEC\ Normal)}$=91, $n_{(UCS\ Tumor)}$=57, $n_{(UCS\ Normal)}$=78. In the box plot, the upper line shows the maxima, the center line shows the median, the lower line shows the minima, the upper bound of box shows upper quartile, the lower bound of box shows lower quartile. Significance was tested using LIMMA. $P_{(BRCA)}$ = 5.53e-39, $P_{(CESC)}$ = 1.74e-10, $P_{(OV)}$ = 6.14e-122, $P_{(PAAD)}$ = 2.67e-63, $P_{(UCEC)}$ = 2.78e-71, $p_{(UCS)}$ = 2.63e-36. *$P$ < 0.01. (B) The relationship between espin expression level and overall survival of patients with tumors in (A). $n_{(Low\ Espin)}$=1098, $n_{(High\ Espin)}$=1098. The 95% confidence interval was added as dotted line. Significance was tested using log-rank test. (C) Diagram showing subcutaneous tumor model in mice. (D) Subcutaneous tumors removed from mice after underarm injection for 10 days. The experiment was independently replicated once. (E) Tumor weights in (D). Data represent technical replicates and are shown as mean ± SD. $n_{(Ctrl)}$ = 7, $n_{(Espin\ OE)}$ = 7. Significance was tested using unpaired Student's $t$ test. (F) Diagram showing experimental tumor metastasis model in mice. (G) Lungs removed from mice after tail-vein injection for 2 weeks. The experiment was independently replicated twice. (H) Quantification of metastatic foci number on the lung surface in (G). Data represent technical replicates and are shown as mean ± SD. $n_{(Ctrl)}$ = 6, $n_{(Espin\ OE)}$ = 6. Significance was tested using unpaired Student's $t$ test. Source data are available online for this figure.

higher espin expression may facilitate tumor metastasis, resulting in lower overall survival.

## Espin has no effect on invadopodia formation or cell survival under fluid shear stress

Cancer metastasis is a multistep process including cancer cell invading from primary tumors, entering circulatory system, surviving and extravasating from the circulation, and colonizing at distant tissues.

Cancer cells utilize invadopodia to traverse basement membranes surrounding primary tumors and invade into adjacent tissues (Murphy and Courtneidge, 2011). To test the effect of espin on invadopodia formation, we performed a fluorescent gelatin degradation assay. Invadopodia were positive for F-actin and cortactin staining, and visible as dark puncta where the fluorescent gelatin was degraded (Fig. 3A). The invadopodia number and the region of degraded gelatin remained consistent between control and espin OE groups, indicating that espin did not affect invadopodia assembly or function (Fig. 3B–D).

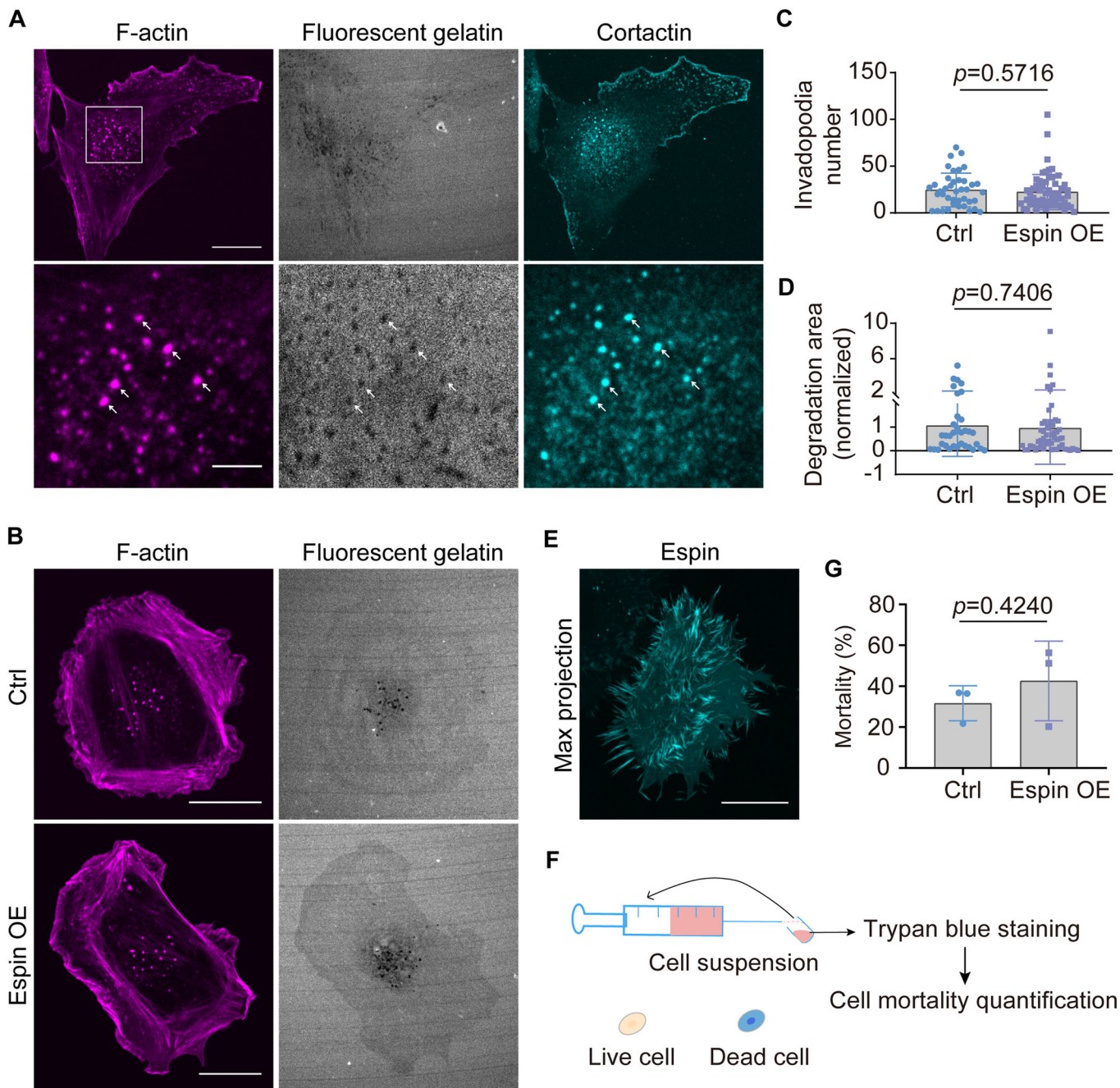

Figure 3.   Espin does not affect invadopodia formation or cell survival under fluid shear stress.

(A) Representative images exhibiting gelatin degradation sites labeled by invadopodia marker actin and cortactin. Arrows mark representative invadopodia. Scale bar: 20 μm in upper image and 5 μm in bottom enlarged image, respectively. (B) Gelatin degradation of control and espin OE cells. Scale bar: 20 μm. (C) Quantification of invadopodia number per cell in (B). Data represent technical replicates and are shown as mean ± SD. $n_{(Ctrl)} = 40$, $n_{(Espin\ OE)} = 61$. Significance was tested using unpaired Student's t test. (D) Quantification of degradation area per cell in (B). Data represent technical replicates and are shown as mean ± SD. $n_{(Ctrl)} = 34$, $n_{(Espin\ OE)} = 55$. Significance was tested using unpaired Student's t test. (E) Representative image of espin OE cell morphology. Scale bar: 20 μm. (F) Schematic diagram of fluid shear stress survival assay. (G) Quantification of mortality of control and espin OE cells after exposure to fluid shear stress. Data represent biological replicates and are shown as mean ± SD. $N_{(Ctrl)} = 3$, $N_{(Espin\ OE)} = 3$. Significance was tested using unpaired Student's t test. The experiment was independently replicated three times. Source data are available online for this figure.

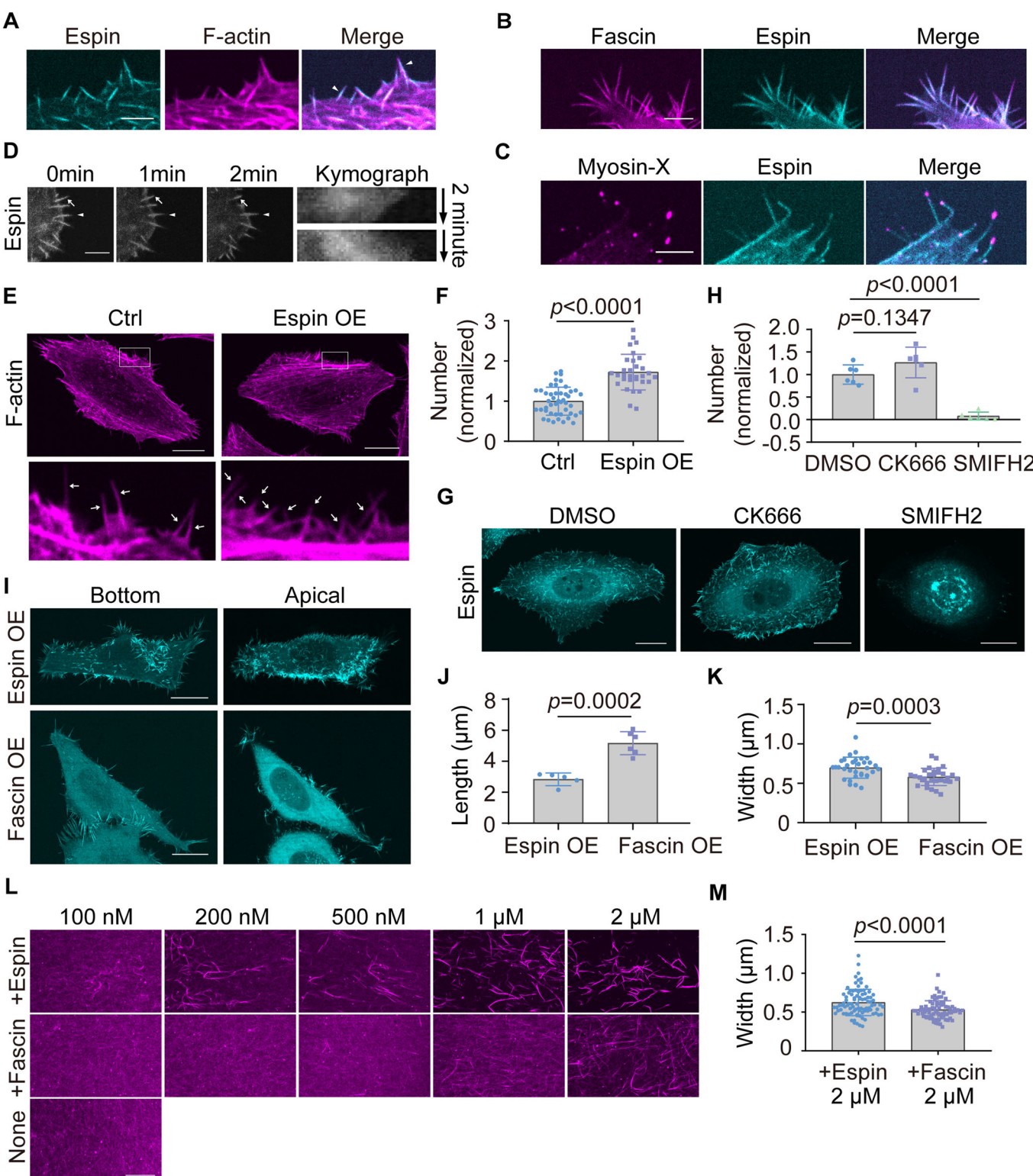

Before reaching distant tissues, cancer cells in the circulation undergo a harsh journey in which less than 0.1% of circulating tumor cells survive due to fluid shear stress (Hope et al, 2021; Regmi et al, 2017). The actin cytoskeleton has been suggested to support cell resistance to fluid shear stress as the depolymerization of F-actin reduced cell viability (Barnes et al, 2012). We noticed the spiky shape of espin OE cells (Fig. 3E) and wondered whether this hairy morphology could enhance fluid shear stress resistance. Using a syringe pump, we applied fluid shear stress to suspended cells and assessed cell mortality after 10 consecutive aspiration and pumping

Figure 4. Espin promotes filopodia formation.

(A) Colocalization of F-actin and espin in cells transfected with espin. Scale bar: 5 µm. (B) Colocalization of fascin and espin on peripheral protrusions. Cells were transfected with fascin and espin. Scale bar: 5 µm. The experiment was independently replicated twice. (C) Localization of myosin-X on the tip of espin-positive protrusions. Cells were transfected with myosin-X and espin. Scale bar: 5 µm. The experiment was independently replicated three times. (D) Dynamic motions of espin decorated filopodia such as extension (triangles) and retraction (arrows). The kymograph above shows retraction while below shows extension. Scale bar: 5 µm. (E) Representative images showing F-actin of control and espin OE cells, arrows mark representative filopodia. Scale bar: 20 µm. The experiment was independently replicated three times. (F) Quantification of filopodia number per cell in (E). Data represent technical replicates and are shown as mean ± SD. $n_{(Ctrl)} = 46$, $n_{(Espin\ OE)} = 31$. Significance was tested using unpaired Student's $t$ test. $P_{(Ctrl,\ Espin\ OE)} = 1.48e\text{-}11$. (G) Representative images showing maximum projection of espin in espin OE cells after treatment of DMSO, CK666 or SMIFH2. Scale bar: 20 µm. The experiment was independently replicated once. (H) Quantification of filopodia number per cell in (G). Data represent technical replicates and are shown as mean ± SD. $n_{(DMSO)} = 6$, $n_{(CK666)} = 6$, $n_{(SMIFH2)} = 6$. Significance was tested using one-way ANOVA and unpaired Student's $t$ test. $P_{(DMSO,\ SMIFH2)} = 1.95e\text{-}6$. (I) Representative images showing espin or fascin in corresponding OE cells. "Bottom" represents the bottom layer while "Apical" represents the maximum projection of dorsal layers. Scale bar: 20 µm. The experiment was independently replicated twice. (J) Quantification of average filopodia length per cell in (I). Data represent technical replicates and are shown as mean ± SD. $n_{(Espin\ OE)} = 5$, $n_{(Fascin\ OE)} = 6$. Significance was tested using unpaired Student's $t$ test. (K) Quantification of filopodia width in (I). Data represent technical replicates and are shown as mean ± SD. $n_{(Espin\ OE)} = 32$, $n_{(Fascin\ OE)} = 30$. Significance was tested using unpaired Student's $t$ test. (L) Representative images showing actin filaments with different concentrations of espin or fascin protein added. Scale bar: 20 µm. The experiment was independently replicated twice. (M) Quantification of F-actin width in (L). Data represent technical replicates and are shown as mean ± SD. $n_{(+Espin\ 2\ µM)} = 89$, $n_{(+Fascin\ 2\ µM)} = 75$. Significance was tested using unpaired Student's $t$ test. $P_{(+Espin\ 2\ µM,\ +Fascin\ 2\ µM)} = 9.13e\text{-}5$. Source data are available online for this figure.

(Fig. 3F) (Barnes et al, 2012). No advantage of espin OE cells was detected in the fluid shear stress survival assay (Fig. 3G). Altogether, the contribution of espin to cancer progression is independent of the modulation of invadopodia formation or cell survival under fluid shear stress.

## Espin exhibits distinct features from fascin in promoting filopodia formation

By expressing EGFP-tagged espin in cells, we observed the localization of espin on cell protrusions which were filled with and dependent on F-actin (Figs. 4A and EV3A,B). When we transfected cells with the canonical filopodial actin-bundling protein fascin or the filopodia tip marker myosin-X (Sousa and Cheney, 2005; Vignjevic et al, 2006), we observed the localization of fascin along the entire length of espin decorated protrusions with myosin-X localized on the tip (Fig. 4B,C), suggesting that these protrusions were filopodia. Espin-positive protrusions showed dynamic extending and retracting like filopodia (Fig. 4D).

Notably, espin OE cells presented a hairy morphology, as reflected by numerous filopodia. Filopodia number quantification revealed a robust role of espin in enhancing filopodia formation (Figs. 4E,F and EV3C). As for the mechanisms of filopodia formation, two alternative models have been presented. The convergent elongation model proposes that filopodial actin filaments are derived from Arp2/3 complex-nucleated actin network, while the de novo filament nucleation is mediated by formins in the absence of Arp2/3 (Evangelista et al, 2001; Svitkina et al, 2003). Respective inhibitors of Arp2/3 or formins were applied to investigate which model espin-enhanced filopodia formation was dependent on. The inhibition of formins, but not Arp2/3, dramatically reduced filopodia number of espin OE cells, suggesting that espin-induced filopodia formation is dependent on formins (Fig. 4G,H).

Fascin, another actin-bundling protein in filopodia, was also shown to positively regulate filopodia number (Lee et al, 2010; Vignjevic et al, 2006). However, espin induced abundant filopodia not only along cell periphery but also towards the dorsal surface while fascin-induced ones mostly distributed at cell periphery but barely at dorsal surface (Figs. 4I and EV3D). Further, espin decorated filopodia were shorter and thicker than fascin-positive ones (Fig. 4J,K), prompting us to investigate the different aspects of

espin and fascin in actin assembly in vitro. As actin-bundling proteins, espin and fascin both showed actin-bundling activity when added to pre-polymerized actin filaments in vitro. Notably, espin exhibited actin-bundling property at the minimum concentration of 100 nM while fascin promoted actin bundling at 500 nM (Fig. 4L). Further, actin filaments were thicker when espin protein was added (Fig. 4M). These observations indicate that espin has stronger actin-bundling activity than fascin, which may endow actin filaments with greater mechanical property.

## Espin contributes to confined cell migration by generating more filopodia

Enhanced confined cell migration can result from increased cell migration speed or ability to squeeze through narrow space. We first evaluated random single-cell motility and found that cell velocity was statistically indistinguishable between espin OE and control groups (Fig. 5A). Wound healing assay of collective migration showed that espin OE cells closed the scratch wound with similar capacity as control cells (Figs. 5B and EV4A), echoing with the single-cell migration result that espin OE may not accelerate cell speed per se.

As shown before, espin positively regulates filopodia formation, which prompted us to explore whether the enhanced confined migration by espin OE was dependent on increased filopodia number. Depletion of filopodia components fascin or myosin-X leads to loss of filopodia (Bohil et al, 2006; Vignjevic et al, 2006). We thus used siRNAs to disrupt fascin or myosin-X expression in espin OE cells and evaluated the effects on confined migration. Of note, deficiency of fascin or myosin-X both induced reduction of transmigrated cells (Figs. 5C,D and EV4B,C), indicating that espin conferred cells with stronger confined migration capacity by generating more filopodia. Specifically, we also generated an espin mutant that lacked the C terminal actin-bundling module (ABM), which was crucial in actin-binding and -bundling (Bartles et al, 1998). Lacking ABM abolished filopodia localization and abrogated filopodia promoting effect of espin (Figs. 5E,F and EV4D,E). Quantification of transmigrated cells revealed that ABM depletion diminished the phenotype of espin-enhanced confined migration (Fig. 5G,H), further supporting that espin contributed to confined migration through promoting filopodia formation which was dependent on its ABM domain.

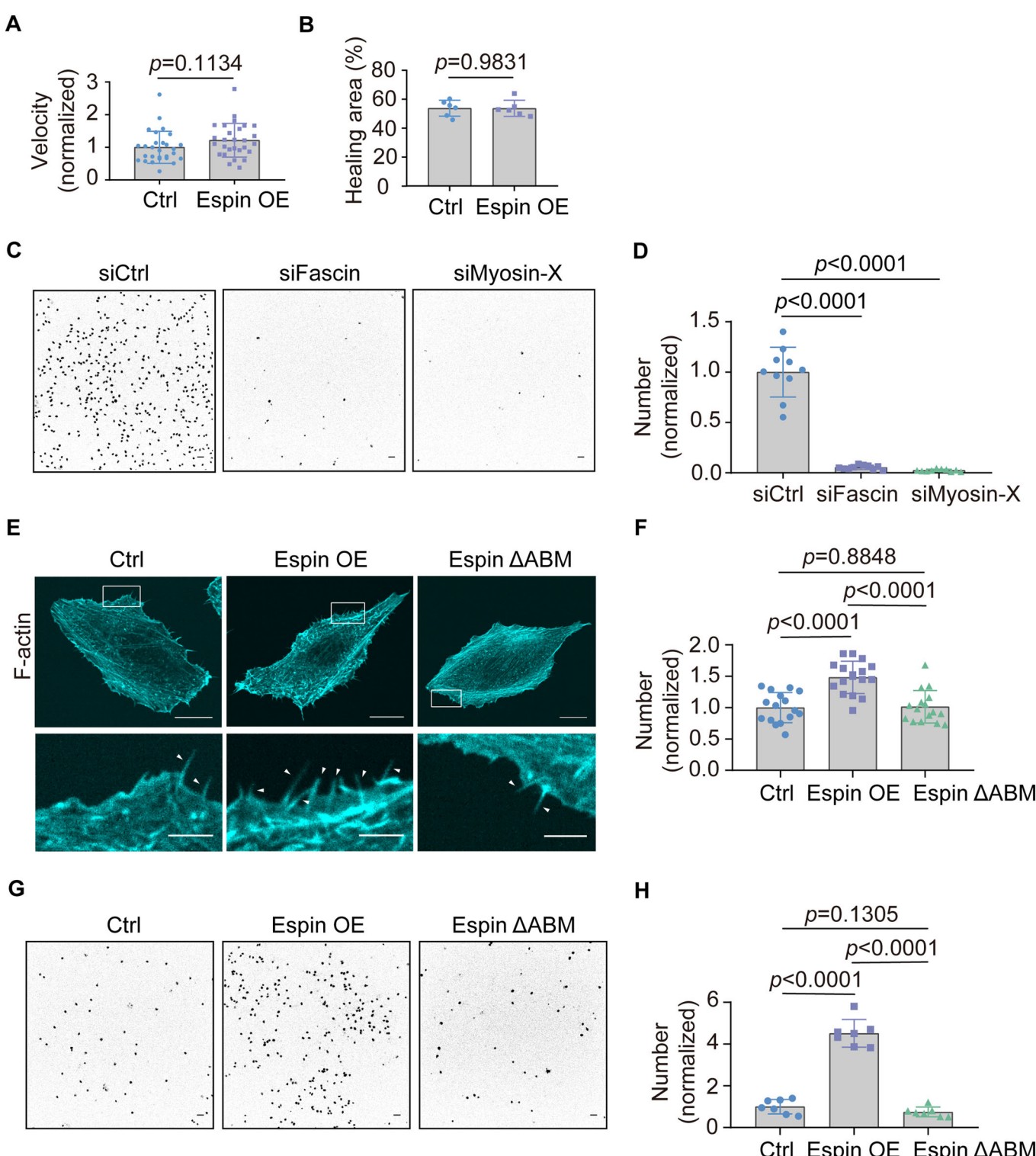

## Espin-rich filopodia form at the leading edge and along the sides in confined cells

To figure out how filopodia contributes to confined cell migration, we imaged cells expressing espin-EGFP migrating in microchannels. Compared with control cells with relatively blunt front, espin-EGFP cells sent out filopodia at the front and along the sides (Fig. 6A). Moreover, we noticed frequent cavities (~40% cells) between the cell body and microchannels at the cell leading part (in front of the nucleus) of espin OE cells, while control cells generally filled the channels fully (Fig. 6B). One explanation for the cavities would be the leading edge of espin OE cells moved faster. We tracked the

**Figure 5. Espin contributes to confined cell migration through promoting filopodia formation.**

(A) Velocity of random single-cell migration on 2D surface. Data represent technical replicates and are shown as mean ± SD. $n_{(Ctrl)} = 28$, $n_{(Espin\ OE)} = 28$. Significance was tested using unpaired Student's $t$ test. The experiment was independently replicated twice. (B) Quantification of relative healing area after 24 h following scratching. Data represent technical replicates and are shown as mean ± SD. $n_{(Ctrl)} = 6$, $n_{(Espin\ OE)} = 6$. Significance was tested using unpaired Student's $t$ test. The experiment was independently replicated once. (C) Representative images of bottom cells on 5 µm transwell, espin OE cells were transfected with siCtrl, siFascin or siMyosin-X. Scale bar: 50 µm. The experiment was independently replicated once. (D) Quantification of cell number in (C). Data represent technical replicates and are shown as mean ± SD. $n_{(siCtrl)} = 10$, $n_{(siFascin)} = 10$, $n_{(siMyosin-X)} = 10$. Significance was tested using one-way ANOVA and unpaired Student's $t$ test. $P_{(siCtrl,\ siFascin)} = 4.77e\text{-}10$, $P_{(siCtrl,\ siMyosin-X)} = 2.76e\text{-}10$. (E) Representative images showing F-actin of cells stably transfected with vehicle, wt espin or espin ΔABM. Cell areas in white boxes are enlarged. Triangles mark representative filopodia. Scale bar: 20 µm in upper images and 5 µm in bottom enlarged images, respectively. The experiment was independently replicated twice. (F) Quantification of filopodia number per cell in (E). Data represent technical replicates and are shown as mean ± SD. $n_{(Ctrl)} = 16$, $n_{(Espin\ OE)} = 17$, $n_{(Espin\ ΔABM)} = 16$. Significance was tested using one-way ANOVA and unpaired Student's $t$ test. $P_{(Ctrl,\ Espin\ OE)} = 4.2e\text{-}6$, $P_{(Espin\ OE,\ Espin\ ΔABM)} = 1.17e\text{-}5$. (G) Representative images of bottom cells on 5 µm transwell, cells were stably transfected with vehicle, wt espin or espin ΔABM. Scale bar: 50 µm. The experiment was independently replicated twice. (H) Cell number quantification in (G). Data represent technical replicates and are shown as mean ± SD. $n_{(Ctrl)} = 7$, $n_{(Espin\ OE)} = 7$, $n_{(Espin\ ΔABM)} = 7$. Significance was tested using one-way ANOVA and unpaired Student's $t$ test. $P_{(Ctrl,\ Espin\ OE)} = 3.23e\text{-}8$, $P_{(Espin\ OE,\ Espin\ ΔABM)} = 7.44e\text{-}9$. Source data are available online for this figure.

leading edge and found that of espin OE cells moved faster than control cells, which may narrow the cell body between the leading edge and the nucleus, resulting in cavities with channels (Fig. 6C).

In confined microchannels, espin also localized at filopodia with fascin and myosin-X labeled (Figs. 6D and EV5A). Notably, we observed that numerous (~0.37 filopodia per µm² at the bottom and ~1.16 filopodia per µm along the sides) and short filopodia (1.58 ± 0.58 µm at the bottom and 0.63 ± 0.21 µm along the sides) of espin OE cells concentrated at the bottom and along the sides (Fig. 6E). These filopodia were devoid of stable focal adhesion signals as shown by tagged paxillin (Figs. 6F and EV5B), though the channels were coated with fibronectin. To map the force distribution in confined channels, we used DNA-based tension probes with pN sensitivity (Wang et al, 2022a) to monitor the tension signals of espin OE cells. The tension signals exhibited partial colocalization with espin-rich filopodia at the leading edge and displayed a sparse dot pattern at the tip of side filopodia (Fig. 6G), indicating that these filopodia may exert force when cells are in confined channels. Although the stable focal adhesions have been reported to be dispensable for cells moving in confined microenvironment, we speculated that the abundant filopodia may integrate with unstable focal adhesions in a specific manner for force generation in confined microchannels. Based on the above observations, we propose that excessive filopodia at the leading edge and along the sides serve for cells to exert force during confined cell migration.

## Discussion

During 2D mesenchymal migration, lamellipodia but not filopodia are indispensable for cell motility. However, excessive filopodia are characteristic of invasive cancer cells and filopodia regulators have long been implicated in cancer metastasis. During metastasis, cancer cells undergo confined migration as they repeatedly encounter mechanical constraints in the microenvironment. Here, we unveiled the role of protrusive structures in facilitating confined migration, specifically the unique role of filopodia, as revealed by espin facilitated confined migration dependent on increased filopodia.

Filopodia were reported to guide microtubules which help nuclear positioning (Koleske, 2003; Renkawitz et al, 2019). In confined channels, espin OE cells formed filopodia at the leading edge, possibly helping position the microtubules frontward the nucleus, pulling the nucleus forward. The integrin-mediated tension signals were detected at the leading edge filopodia. These filopodia may integrate with integrin and generate force for

migration forward. Aside from the leading edge, filopodia also concentrate along the sides and at the bottom. During 2D mesenchymal migration, cells extend filopodia not only along the edges but also towards the dorsal side (Bohil et al, 2006), whose function is barely studied. In 3D microenvironment, extensive filopodia distribute along cell surface, possibly serving as contacts with the surrounding microenvironment and contributing to 3D migration.

It has been reported that focal adhesions are dispensable for cells moving in confined microenvironment. Stable matrix adhesions are absent and exert less force when cells are in confined microchannels instead of wider channels (Holle et al, 2019; Wang et al, 2022a). In confined channels, we noticed that numerous and short filopodia along the sides and at the bottom were devoid of stable focal adhesions. Sparse tension signals decorated at the tip of some of these filopodia, supporting that the force generation herein and that extensive filopodia may provide force for cells during confined migration. Cell-substrate intercalation model has been proposed for force transmission between cell and substrate during migration without focal adhesions (Paluch et al, 2016). Protrusions are extended when cancer cells are in trans-endothelial migration or during lymphocytes migrating through 3D matrix (Haston et al, 1982; Herman et al, 2019; Jacquemet et al, 2015). Side filopodia may function to intercalate into surrounding matrix or endothelium, thereby contributing to efficient confined migration of cancer cells. The interaction between filopodia and dense matrix or endothelial cells remains to be explored and may provide new insights into cancer progression.

## Methods

**Reagents and tools table**

| Reagent/ resource | Reference or source | Identifier or catalog number |
|---|---|---|
| **Experimental models** | | |
| B16-F10 | Fuping You Laboratory (Peking University, China) | |
| C57BL/6J (M. musculus) | Animal Center of Peking University Health Science Center | |
| HEK293T | Yuxin Yin Laboratory (Peking University, China) | |

| Reagent/ resource | Reference or source | Identifier or catalog number |
|---|---|---|
| MDA-MB-231 | Yujie Sun Laboratory (Peking University, China) | |
| MEF | James Bear Laboratory (University of North Carolina at Chapel Hill, USA) | |
| **Recombinant DNA** | | |
| GFP-myosin-X | Huali Yu Laboratory (Northeast Normal University, China) | |
| mEmerald-Espin-C | Addgene | #54084 |
| pLKO.1 | Addgene | #8453 |
| pLVX-AcGFP-fascin | cDNA library derived from MDA-MB-231 cells | |
| pLVX-paxillin-mScarlet | cDNA library derived from MDA-MB-231 cells | |
| **Antibodies** | | |
| Mouse anti-α-tubulin | Abclonal | AC012 |
| Rabbit anti-cortactin | Temecula, CA | 05-180 |
| Rabbit anti-espin | Abclonal | A15908 |
| Rabbit anti-fascin | Abclonal | A9566 |
| Rabbit anti-myosin-X | Abclonal | A12466 |
| **Oligonucleotides and other sequence-based reagents** | | |
| Espin forward primer | This study | 5′- ATGGCCCTGGAGCAGGC-3′ |
| Espin reverse primer | This study | 5′- GTACTTAGCGATGTCCCCCT-3′ |
| Fascin forward primer | This study | 5′-ATGACCGCCAACGGCACAG-3′ |
| Fascin reverse primer | This study | 5′-GTACTCCCAGAGCGAGGCGG-3′ |
| Paxillin forward primer | This study | 5′-ATGGACGACCTCGACGCCC-3′ |
| Paxillin reverse primer | This study | 5′-GCAGAAGAGCTTGAGGAAGCAG-3′ |
| shCtrl | This study | 5′-AACGCTGCTTCTTCTTATTTA-3′ |
| shEspin-1 | *Homo sapiens* | 5′-GAGCTACATGGACATGCTGAA-3′ |
| shEspin-2 | *Homo sapiens* | 5′-CCAAGTCTTTCAACATGATGT-3′ |
| siCtrl | Sangon Biotech | 5′-UUCUCCGAACGUGUCACGUTT-3′ |
| siFascin | Sangon Biotech | 5′-GAGCAUGGCUUCAUCGGCUTT-3′ |
| siMyosin-X | Sangon Biotech | 5′-AAGUGCGAACGGCAAAAGAGA-3′ |
| **Chemicals, enzymes, and other reagents** | | |
| 2% gelatin | Sciencell | 0423 |
| Alexa FluorTM Plus 647 Phalloidin | Invitrogen | A30107 |
| CCK8 | GLPBIO | GK10001 |
| CK666 | Sigma | 182515 |
| Escherichia coli BL21-CondonPlus (DE3)-RIPL | AngYu Biotechnologies | AYBIO-G6028 |
| Hoechst 34580 | Thermo | H21486 |

| Reagent/ resource | Reference or source | Identifier or catalog number |
|---|---|---|
| Latrunculin B | Sigma | L5288 |
| NeofectTM | Fengmao Technology | TF20121201 |
| opti-MEM | Invitrogen | 31985-070 |
| Oregon Green 488 conjugated gelatin | Invitrogen | G13186 |
| PDMS | Momentive Performance Materials | RTV615 |
| ProLongTM Glass Antifade Mountant with NucBlueTM Stain | Invitrogen | P36981 |
| Rhodamine-labeled rabbit skeletal muscle actin | Cytoskeleton | AR05 |
| SMIFH2 | Sigma | S4826 |
| SYBR Green | ABclonal | RK21203 |
| Transcript One-Step gDNA Removal and cDNA Synthesis SuperMix Kit | Transgene | AT311-02 |
| Transwell (5 μm) | Corning, Costar | #3421 |
| Trizol | Life Technologies | 15596026 |
| Trypan Blue | Sigma | T8154 |
| **Software** | | |
| Graph Pad Prism 8 | | |
| ImageJ | | |
| **Other** | | |

## Cell culture and drug treatment

MEF cells and MDA-MB-231 cells were generously provided by James Bear laboratory (University of North Carolina at Chapel Hill, USA) and Yujie Sun laboratory (Peking University, China), respectively. Human embryonic kidney 293T (HEK293T) cells were a generous gift from Yuxin Yin laboratory (Peking University, China). B16-F10 cells were generously provided by Fuping You laboratory (Peking University, China). MEF, MDA-MB-231, and HEK293T cells were cultured in Dulbecco's modified Eagle medium (DMEM; Corning, 10-013-CRVC) supplemented with 10% fetal bovine serum (FBS; PAN-Biotech, P30-3302), 100 U/mL penicillin and 100 mg/mL streptomycin at 37°C with 5% $CO_2$. B16-F10 cells were maintained in Roswell Park Memorial Institute (RPMI) 1640 medium supplemented with 10% FBS, 100 U/mL penicillin, and 100 mg/mL streptomycin at 37°C with 5% $CO_2$.

Latrunculin B (Sigma, L5288) was used at a concentration of 250 nM to induce F-actin depolymerization. CK666 (Sigma, 182515) and SMIFH2 (Sigma, S4826) were used at concentrations of 100 μM and 25 μM, respectively, for 5 h to inhibit Arp2/3 complex or formin.

## Plasmid construction and transfection

Espin was cloned from mEmerald-Espin-C (Addgene, #54084) and subcloned into a lentiviral vector (pLVX-AcGFP-N1) with C terminal

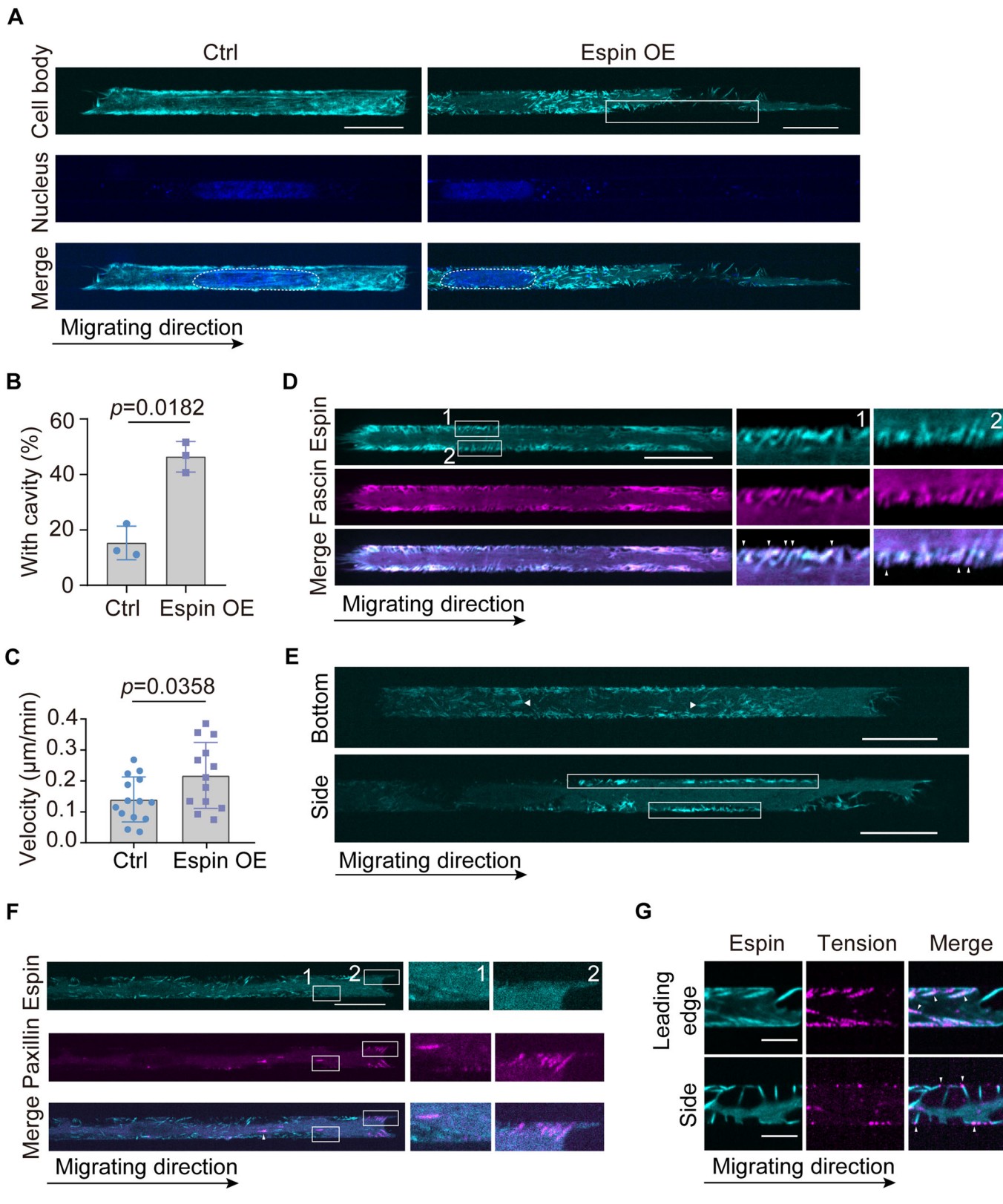

Figure 6.  Espin-rich filopodia form at the leading edge and along the sides in confined cells.

(A) Representative images of control and espin OE cells in confined microchannels with 5 μm width. Images left show control cell labeled by LifeAct while right images show espin-EGFP cell. The nuclei were highlighted with dashed lines in merge images. The rectangle marks the cavity between cell body and the channel. Scale bar: 20 μm. The experiment was independently replicated three times. (B) Ratio quantification of cells forming cavities with the microchannels. Data represent biological replicates and are shown as mean ± SD. $N_{(Ctrl)} = 3$, $N_{(Espin\ OE)} = 3$. Significance was tested using unpaired Student's t test. The experiment was independently replicated three times. (C) Velocity quantification of the leading edge. Data represent technical replicates and are shown as mean ± SD. $n_{(Ctrl)} = 14$, $n_{(Espin\ OE)} = 13$. Significance was tested using unpaired Student's t test. The experiment was independently replicated once. (D) Representative images of fascin colocalization with espin in confined channels. Espin OE cells were transfected with fascin. Cell areas in white boxes are enlarged. Triangles mark representative filopodia with espin and fascin colocalized. Scale bar: 20 μm. The experiment was independently replicated twice. (E) Representative images of espin OE cells displaying abundant filopodia at the bottom and along the sides, marked with triangles and rectangles, respectively. Scale bar: 20 μm. (F) Representative images of focal adhesions in espin OE cells in confined channels. Espin OE cells were transfected with paxillin. Scale bar: 20 μm. The experiment was independently replicated twice. (G) Representative images of espin OE cells in microchannels coated with 17 pN DNA probes. The upper images show the tension signals on filopodia at the leading edge while the bottom images show the tension signals at the tip of side filopodia. Triangles mark representative tension signals along the leading edge filopodia or at the tip of side filopodia. Scale bar: 5 μm. The experiment was independently replicated three times. Source data are available online for this figure.

tag EGFP. The espin ΔABM truncation was amplified and cloned into the same vector. The human fascin was cloned from a cDNA library derived from MDA-MB-231 cells and tagged with AcGFP at N terminus. The human paxillin was cloned from a cDNA library derived from MDA-MB-231 cells and tagged with mScarlet at C terminus. The GFP-myosin-X construct was a generous gift from Huali Yu laboratory (Northeast Normal University, China).

For DNA transfection, cells were transfected with 2 μg plasmid DNA in opti-MEM (Invitrogen, 31985-070) containing 2 μL NeofectTM DNA transfection reagent (TF20121201) following the protocol for 24–48 h.

For espin knockdown in MDA-MB-231 cells, the following shRNAs were cloned into pLKO.1 vector (Addgene, #8453). After lentivirus infection and puromycin selection, the protein levels of pooled cells were validated by western blotting.

shCtrl: 5'-AACGCTGCTTCTTCTTATTTA-3';
shEspin-1: 5'-GAGCTACATGGACATGCTGAA-3';
shEspin-2: 5'-CCAAGTCTTTCAACATGATGT-3'.

For fascin and myosin-X knockdown in MDA-MB-231 cells, the following siRNAs were synthesized by Sangon Biotech. After 48 h following siRNA treatment, the cells were harvested to determine the protein levels or used for transwell migration assay.

siCtrl: 5'-UUCUCCGAACGUGUCACGUTT-3';
siFascin: 5'-GAGCAUGGCUUCAUCGGCUTT-3';
siMyosin-X: 5'-AAGUGCGAACGGCAAAAGAGA-3'.

## Real-time quantitative PCR

The cells were lysed using Trizol (Life Technologies, 15596026) and total RNA was extracted with chloroform and methanol. RNA was reverse transcribed using a Transcript One-Step gDNA Removal and cDNA Synthesis SuperMix Kit (Transgene, AT311-02). The levels of the ESPN and ACTB mRNA were amplified and analyzed using SYBR Green (ABclonal, RK21203). Data shown are the relative abundance of the ESPN mRNA normalized to ACTB.

## RNA sequencing (RNA-seq)

RNA extraction and sequencing were performed by Berry Genomics Co., Ltd (http://www.berrygenomics.com/. Beijing, China). Total RNA was extracted using Trizol Reagent (Invitrogen, 15596018). Total RNA was qualified and quantified as follows: (1) RNA purity and concentration were examined using NanoDrop 2000; (2) RNA integrity and quantity were measured using the Agilent 2100/4200 system.

Then, mRNA was purified from total RNA using polyT and then fragmented into 300–350 bp fragments. The first strand cDNA was reverse transcribed using fragmented RNA and dNTPs (dATP, dTTP, dCTP and dGTP) and second strand cDNA was synthesized using DNA polymerase I and dNTPs (dATP, dUTP, dCTP and dGTP). The remaining overhangs of double-strand cDNA were converted into blunt ends via exonuclease/polymerase activities. After adenylation of 3' ends of DNA fragments, sequencing adapters were ligated to the cDNA and the library fragments were purified. The template without U was enriched by PCR, and the PCR product was purified to obtain the final library. After library construction, the concentration of the library was measured by the Qubit® fluorometer. The accurate concentration of cDNA library was again examined using qPCR. The size distribution of the library was detected by agarose gel electrophoresis. After library preparation and pooling of different samples, the samples were subjected for NGS platform by Berry Genomics Co., Ltd (http://www.berrygenomics.com/, Beijing, China).

## Western blotting

For western blotting, cells were washed with phosphate-buffered saline (PBS) twice and lysed with appropriate volumes of RIPA buffer (50 mM Tris-HCl, pH 8.0, 150 mM NaCl, 1% Triton X-100, 0.5% Na-deoxycholate, 0.1% SDS, 1 mM EDTA and protease inhibitor cocktail) for 15 min on ice. The cell lysis was centrifuged at 12000 rpm for 10 min at 4°C, and the supernatants were collected. Then, 5×SDS loading buffer was added to the supernatants, and the mixtures were boiled for 10 min at 95°C. Protein samples were run on 6% or 10% SDS-PAGE acrylamide gels and transferred onto nitrocellulose (NC) membranes by wet electrophoretic transfer, followed by blocking (5% skim milk) and primary antibody incubation at 4°C overnight: rabbit anti-espin (Abclonal, A15908), rabbit anti-fascin (Abclonal, A9566), rabbit anti-myosin-X (Abclonal, A12466) and mouse anti-α-tubulin (Abclonal, AC012). After washed with PBST (PBS with 2‰ Tween-20) three times, the membranes were incubated with secondary antibody at room temperature (RT) for 1 h and then washed with PBST three times. Blots were visualized and recorded using SH-Compact 523 (SHST).

## Immunofluorescence and imaging

Cells grown on coverslips were fixed with 4% paraformaldehyde at RT for 15 min, permeabilized in 0.5% Triton X-100 in PBS for 5 min, and then blocked with 10% bovine serum albumin (BSA) for

1 h. The primary antibodies were incubated at RT for 1 h: rabbit anti-cortactin (Temecula, CA, 05-180). After washed with PBS three times, the coverslips were incubated with Alexa Fluor 488 or 555-conjugated secondary antibody at RT for 1 h. F-actin was stained with Alexa Fluor™ Plus 647 Phalloidin (Invitrogen, A30107). Then the coverslips were washed with PBS three times and mounted with ProLong™ Glass Antifade Mountant with NucBlue™ Stain (Invitrogen, P36981). After mounting medium was solidified, images were captured by Andor Dragonfly confocal imaging system or Leica STELLARIS 8 with stimulated emission depletion (STED).

### Transwell migration assay

Transwell migration assay was performed using 24-well inserts with 5 µm size pores (10 µm long) in polycarbonate membranes (Corning, Costar #3421). In total, $2.5 \times 10^5$ cells were seeded on the top of the insert in 250 µL medium without serum. Complete medium with serum was added to the bottom. The insert was kept incubated at 37 °C and 5% $CO_2$. After 16 h, cells were stained with Hoechst 34580 (Thermo, H21486) at 2.5 µg/mL for 20 min. The non-migrated cells present on the upper surface were removed with a cotton swab. The migrated cells on the bottom were imaged for cell number quantification.

### Confined microchannel fabrication

The confined microchannel was fabricated as previously published (Wang et al, 2022b). A mold is needed to imprint the structures on the confined microchannels. This mold can be produced by photolithography of a layer of photoresist (SU-8, Kayaku Advanced Materials) deposited on a silicon wafer by using standard microfabrication protocols. Afterward, the mold was covered with PDMS (Momentive Performance Materials, RTV615) in the 10:1 ratio of base versus curing agent. Then bake the mold with coverslips on the hot plate at 80°C for 15 min, and remove the PDMS rim with tweezers. Use a razor blade to gently unstick the channel from the mold. The glass bottoms of confocal dishes were cleaned with isopropanol. After drying up, the glass bottom and confinement channels were exposed to plasma for 3 min, bonded with each other, and then incubated with 20 µg/mL fibronectin at 4°C overnight. They were rinsed twice with PBS, and incubated with medium for 30 min. In total, $2 \times 10^6$ cells were seeded in the sample hole and imaged for trajectory tracking or filopodia observing.

### Integration of microchannel with DNA probes

The reversible shearing DNA-based tension probes and gold nanoparticles coated coverslips were prepared as described as previously published (Wang et al, 2022a). Coverslips and microchannels were bonded together and then coated with DNA probes (200 nM) at RT for 1 h.

### Wound healing assay

Cells were seeded in 6-well culture plates until the confluence reached 100% and then scraped in a straight line with a sterile pipette tip. PBS was used to wash out suspended cells. The distances at onset and that of cells had migrated for 24 h were photographed at the same position using the Olympus-inverted microscope. ImageJ Plugin Measure Wound Healing Coherency was used to measure and calculate the gap area that the cells had migrated.

### Cell proliferation assay

The effect of espin on cell proliferation was assessed by CCK8 (GLPBIO, GK10001) according to the manufacturer's instructions. In brief, cells were suspended in culture medium and inoculated in 96-well plates ($5 \times 10^3$ cells/well, 100 µL). After 24, 48 or 72 h incubation, 10 µL CCK8 solution was added to each well. The plate was incubated at 37°C for an additional 1 h before measuring the absorbance at 450 nm wavelength.

### Fluorescent gelatin degradation assay

For fluorescent gelatin coating, sterilized glass coverslips were incubated in 50 µg/mL poly-L-lysine for 20 min at RT. Following incubation, coverslips were rinsed three times in PBS and then incubated in ice-cold 0.5% glutaraldehyde for 15 min at 4°C. Unlabeled 2% gelatin (Sciencell, 0423) was mixed with Oregon Green 488 conjugated gelatin (Invitrogen, G13186) at 8:1 ratio after allowing gelatin to fully dissolve in a 37°C water bath. A drop (35 µL) of fluorescent gelatin mixture was deposited on a flat parafilm, and coverslips were inverted onto the drop with its poly-L-lysine- and glutaraldehyde-coated surface facing down, and incubated for 30 min at RT in the dark. After being washed with PBS for three times, coverslips were used for gelatin degradation in medium at 37°C and processed for immunofluorescence.

### Fluid shear stress assay

The cell suspension was expelled by a syringe pump through a needle (0.45×16 RWLB). The entire contents of the syringe were passed through the needle (considered one exposure) and collected in a 5 mL tube. The process was repeated ten times by drawing the suspension into a needle-less syringe. Prior to the first exposure (non-FSS exposed control) and after ten exposures, 100 µL aliquots were removed from the collected suspension and used for mortality measurement using Trypan Blue (Sigma, T8154) staining.

### Protein purification and actin-bundling assay

For recombinant protein expression in Escherichia coli BL21-CondonPlus (DE3)-RIPL (AngYu Biotechnologies, AYBIO-G6028), DNA fragments encoding espin or fascin were cloned into the modified pET28a (+) vector. When bacteria were cultured at 37°C to an $OD_{600}$ of 0.8, protein expression was induced with 0.1 mM isopropyl-1-thio-β-D-galactopyranoside (IPTG) for 16 h at 18°C. For protein purification, cell pellets were collected by centrifugation at 4000 rpm and suspended in TBS buffer containing 30 mM Tris (pH 8.0), 140 mM NaCl, 3 mM KCl, 10% glycerol, 0.5 mM TCEP, and 1 mM PMSF. The suspension was lysed by sonication at 4°C and cleared by centrifugation at 4°C, 13,000 rpm for 30 min. The supernatant was applied to a Ni-IDA column and washed three times using buffers containing 20 mM Tris (pH 8.0), 150 mM NaCl, 0.5 mM TCEP, and different concentrations of imidazole (0 mM,

20 mM, 40 mM). Proteins were eluted with 300 mM imidazole and purified by size exclusion chromatography using Superdex200 Increase 10/300 GL.

For actin-bundling assay, rhodamine-labeled rabbit skeletal muscle actin (3 µM; Cytoskeleton, AR05) was polymerized at RT for 0.5 h in buffer containing 5 mM Tris-HCl (pH 8.0), 50 mM KCl, 5 mM MgCl$_2$, 1 mM EGTA, 10 mM imidazole, 200 µM ATP and 0.5 mM DTT. Then, actin was mixed with different concentrations of espin or fascin and imaged using Andor Dragonfly confocal imaging system by putting a flow chamber upside down on the objective. Chambers were prepared as follows. Glass slides and coverslips were rinsed with ethanol and dried up. The chamber with two openings on opposite sides was then made by attaching the coverslip to the glass slide using double-sided tape placed parallel to each other.

## Scanning electron microscopy (SEM)

Cells were seeded on the ACLAR® Films (50425) in 12-well plates and cultured for 12 h before fixation. Then cells were washed with 37°C PB buffer (0.2 M NaH$_2$PO$_4$ and Na$_2$HPO$_4$, pH 7.4) mixed with isopycnic DMEM for once, and immediately fixed with PB buffer containing 2.5% glutaraldehyde at RT for 1 h. Images were acquired using Quanta FEG 450 electron microscope.

## Mice tumor model

Adult male C57BL/6J mice (5–6 weeks) were provided by the Animal Center of Peking University Health Science Center (Beijing, China). All animals care and use adhered to the Guide for the Care and Use of Laboratory Animals of the Chinese Association for Laboratory Animal Science, and mice were housed under 12/12 h light/dark cycle. No subjective bias during animal treatment. The University Animal Care Committee for Animal Research of Peking University Health Science Center approved the study protocol (approval code BCJB0010).

For the subcutaneous tumor model, mice were subcutaneously injected in the left flank with 1×10$^6$ B16-F10 cells in 100 µL PBS. After 10 days, animals were euthanized. The subcutaneous tumors were removed and weighed. For the experimental lung metastasis model, mice were intravenously injected in the tail vein with 1×10$^6$ B16-F10 cells in 100 µL PBS. After 2 weeks, animals were euthanized, and their lungs were removed and rinsed in PBS for measuring metastatic foci.

## Quantification and statistical analysis

In filopodia number quantification, we included protrusions labeled by F-actin. In filopodial length quantification, the filopodia were detected from espin or fascin channel. The filopodia origins from where the intensity was above the cytoplasm and reaches where the intensity was above the background. The filopodial length was calculated as the distance between the two points. The widths of F-actin were defined as the length with fluorescence intensity above the background along lines perpendicular to the direction of F-actin.

The volcano plot was acquired using NetworkAnalyst (Zhou et al, 2019). The expression profiles of espin among tumor samples and normal samples were obtained using Gene Expression Profiling Interactive Analysis (GEPIA) (Tang et al, 2017). The overall survival of patients with tumors was also obtained from GEPIA. The group cutoff showed median. The synopsis was created by Biorender.

Unpaired Student's *t* test and LIMMA was used to assess the significance when comparing only two groups. For experiments comprising more than two groups, one-way ANOVA test was used. No blinding was done.

## Data availability

The mRNA-seq data have been deposited to the GEO (Gene Expression Omnibus) database on NCBI and assigned the accession no. GSE286976.

The source data of this paper are collected in the following database record: biostudies:S-SCDT-10_1038-S44319-025-00437-1.

## Peer review information

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

## Acknowledgements

The authors thank Dr. Wenzhong Yang, Dr. Xiaoyu Ren and Dr. Xin Yi from Peking University for helpful discussion. The authors thank Dr. Huali Yu (Northeast Normal University, China) for the gift of GFP-myosin-X plasmid. The authors thank the National Center for Protein Sciences and Core Facilities of Life Sciences at Peking University for assistance with SEM sample and the authors would be grateful to thank Yiqun Liu, Pengyuan Dong and Hongmei Zhang for their help with SEM. The authors thank Tsinghua-Peking Center for Life Sciences for assistance with STED and the authors would be grateful to thank Bingyu Liu for the help with STED. This work was supported by funding from the National Key R&D Program of China (2022YFC3401100) and National Natural Science Foundation of China 32122029 for Congying Wu, 32300635 for Peng Shi; China Postdoctoral Science Foundation (2022M710263) for Peng Shi.

## Author contributions

**Yan Wang**: Conceptualization; Formal analysis; Investigation; Methodology. **Peng Shi**: Conceptualization; Funding acquisition; Investigation; Writing—review and editing. **Geyao Liu**: Investigation; Visualization. **Wei Chen**: Investigation. **Ya-Jun Wang**: Investigation; Visualization; Writing—original draft. **Yiping Hu**: Formal analysis; Investigation. **Ao Yang**: Investigation. **Tonghua Wei**: Formal analysis; Investigation. **Yu-Chen Chen**: Investigation. **Ling Liang**: Investigation. **Zheng Liu**: Conceptualization; Investigation. **Yan-Jun Liu**: Writing—original draft. **Congying Wu**: Conceptualization; Funding acquisition; Writing—original draft; Writing—review and editing.

Source data underlying figure panels in this paper may have individual authorship assigned. Where available, figure panel/source data authorship is listed in the following database record: biostudies:S-SCDT-10_1038-S44319-025-00437-1.

## Disclosure and competing interests statement

The authors declare no competing interests.

# Expanded View Figures

**Figure EV1.   Related to Fig. 1.**

(**A**) Confocal z-stacks taken with 1 µm steps show that the constriction length of transwell is approximately 10 µm. Stack 1 shows bottom nuclei on transwell while stack 12 shows upper nuclei. Dashed circles mark the representative bottom and upper nuclei. Scale bar: 20 µm. (**B**) Western blotting showing espin KD. (**C**) Cell growth curve of control and espin OE cells using CCK8 kit. Data represent technical replicates and are shown as mean ± SD. $n_{(Ctrl)} = 5$, $n_{(Espin\ OE)} = 5$. (**D**) Cell growth curve of control and espin KD cells. Data represent technical replicates and are shown as mean ± SD. $n_{(shCtrl)} = 5$, $n_{(shEspin-1)} = 5$, $n_{(shEspin-2)} = 5$. (**E**) Representative images showing control (just blue) and espin OE cells (cyan and blue) in microchannels. The nuclei were stained with Hoechst and displayed as blue. Scale bar: 20 µm. (**F**) Representative nuclear displacement of control and espin OE cells in confined microchannels with 5 µm width. The nuclei were stained with Hoechst and displayed as blue. Scale bar: 20 µm.

▶

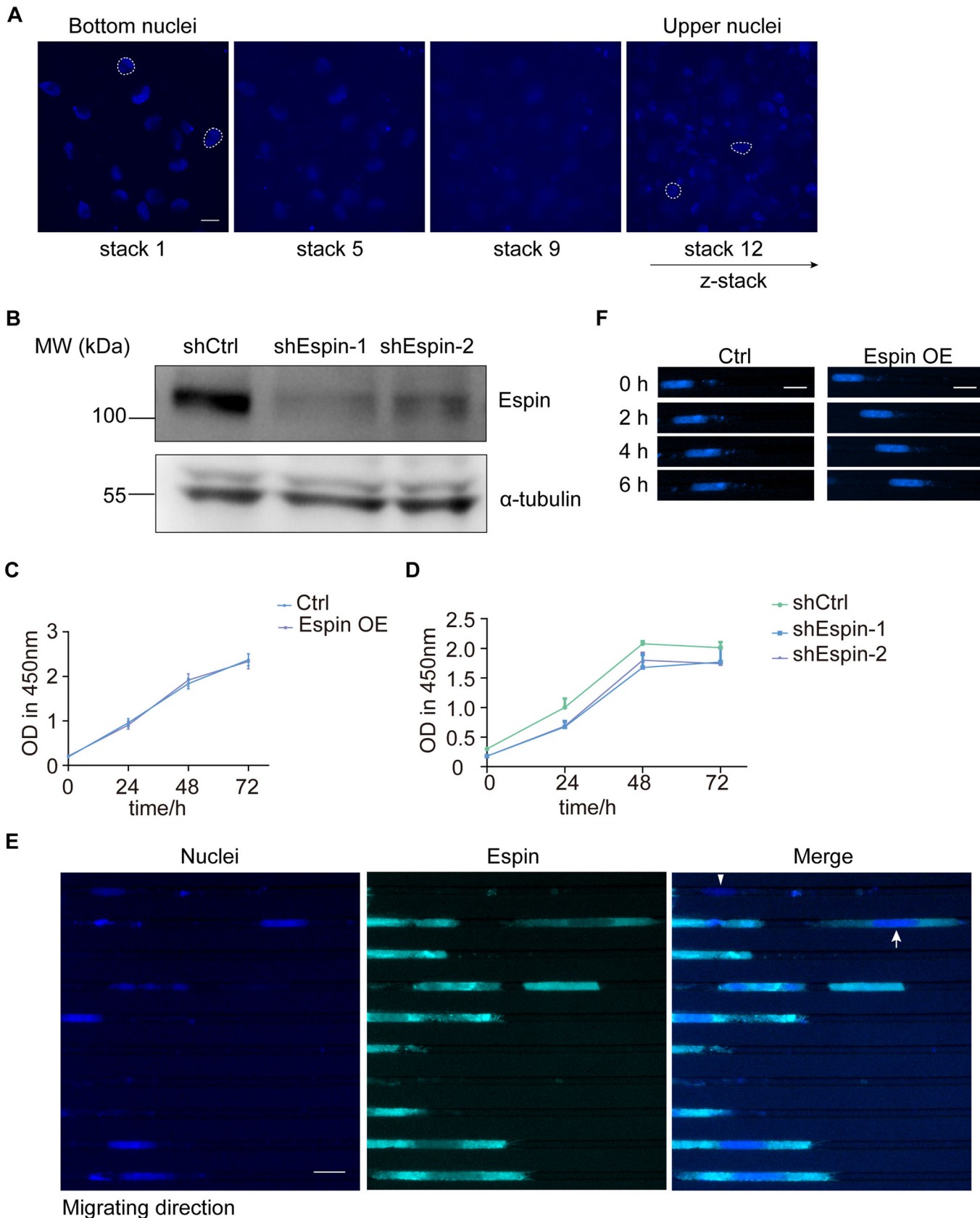

**A**

Bottom nuclei

Upper nuclei

stack 1 stack 5 stack 9 stack 12

z-stack

**B**

MW (kDa) shCtrl shEspin-1 shEspin-2

100 —

Espin

55 —

α-tubulin

**F**

Ctrl Espin OE

0 h

2 h

4 h

6 h

**C**

OD in 450nm

— Ctrl
— Espin OE

time/h

**D**

OD in 450nm

— shCtrl
— shEspin-1
— shEspin-2

time/h

**E**

Nuclei Espin Merge

Migrating direction

**A**

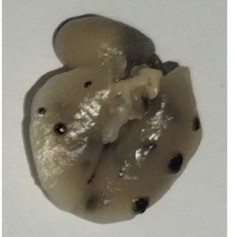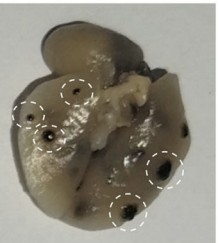

**Figure EV2.   Related to Fig. 2.**

(**A**) Metastatic foci are morphologically circular and black to be visible on the lung surface, as shown in dashed circles.

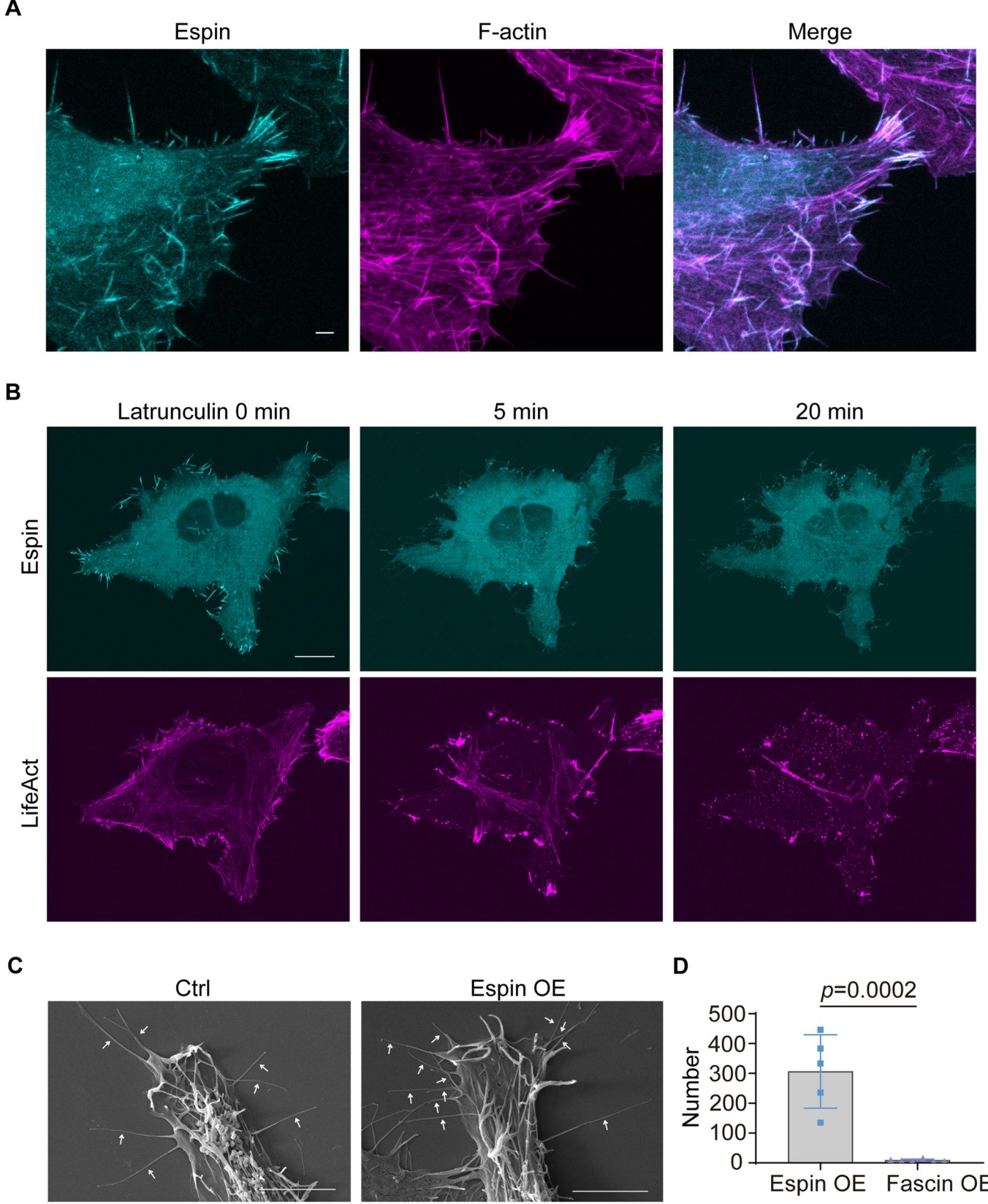

**A**

Espin | F-actin | Merge

**B**

Latrunculin 0 min | 5 min | 20 min

Espin

LifeAct

**C**

Ctrl | Espin OE

**D**

*p*=0.0002

◀

**Figure EV3. Related to Fig. 4.**

(A) Representative images of F-actin staining and espin-EGFP using STED. Scale bar: 5 μm. (B) Signals of LifeAct and espin. Cells transfected with espin and LifeAct were treated with 250 nM latrunculin B. Scale bar: 20 μm. (C) Representative images of cell morphology using SEM, triangles mark filopodia. Scale bar: 5 μm. (D) Quantification of dorsal filopodia number per cell in Fig. 4I. Data represent technical replicates and are shown as mean ± SD. $n_{(Espin\ OE)} = 5$, $n_{(Fascin\ OE)} = 6$. Significance was tested using unpaired Student's $t$ test.

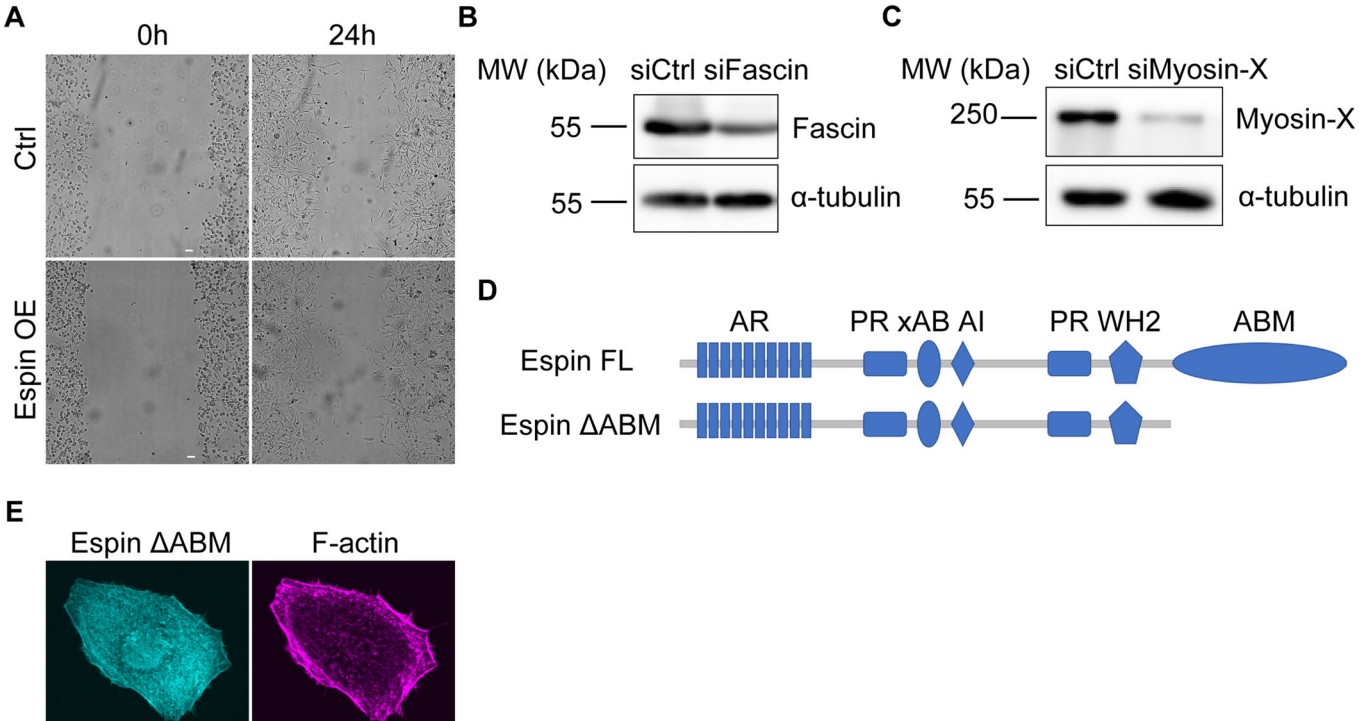

**Figure EV4.   Related to Fig. 5.**

(A) Representative images showing the initial and 24 h area in wound healing assay. Scale bar: 50 µm. (B) Western blotting showing fascin KD in espin OE cells. Due to the similar molecular weights of fascin and α-tubulin, siCtrl and siFascin samples were loaded twice on the same 10% SDS-PAGE acrylamide gel. (C) Western blotting showing myosin-X KD in espin OE cells. (D) The structure of espin full length (FL) and the mutant depleting ABM. (E) Fluorescent signals of espin ΔABM and F-actin. Scale bar: 20 µm.

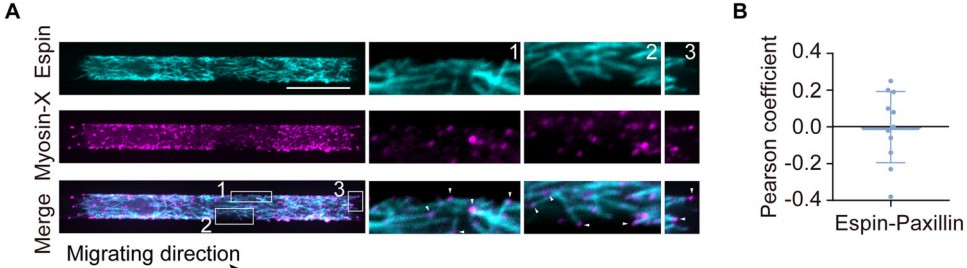

**Figure EV5.   Related to Fig. 6.**

(A) Representative images of myosin-X localization and espin in confined channels. Espin OE cells were transfected with myosin-X. Cell areas in white boxes are enlarged. Triangles mark representative myosin-X at the tip of espin-protrusions. Scale bar: 20 µm. (B) The Pearson coefficient of espin and paxillin in confined cells. To filter unspecific background signals, a manual intensity threshold was used when using the Coloc2 in ImageJ to calculate Pearson coefficients. Data represent technical replicates and are shown as mean ± SD. $n_{(Espin-Paxillin)} = 11$.

                                  