## [Peer Review File · EMBO Reports]

Espin enhances confined migration by inducing filopodia formation and promotes cancer metastasis

Yan Wang, Peng Shi, Geyao Liu, Wei Chen, Ya-Jun Wang, Yiping Hu, Ao Yang, Tonghua Wei, Yu-Chen Chen, Ling Liang, Zheng Liu, Yan-Jun LIU, and Congying Wu

Corresponding author(s): Congying Wu (congyingwu@hsc.pku.edu.cn), Zheng Liu (zheng.liu@whu.edu.cn), Yan-Jun LIU (Yanjun_Liu@fudan.edu.cn), Peng Shi (pengshi@suda.edu.cn)

Review Timeline:

Submission Date:	24th Sep 24
Editorial Decision:	21st Oct 24
Revision Received:	8th Jan 25
Editorial Decision:	18th Feb 25
Revision Received:	22nd Feb 25
Accepted:	7th Mar 25

Editor: Deniz Senyilmaz Tiebe

Transaction Report:

Dear Congying,

Thank you for submitting your research manuscript to our journal, which was now seen by two referees, whose reports are copied below.

Referees express interest in the proposed role of espin in confined cell migration. However, they, especially referee #1, also raises concerns that need to be addressed for publication here. In particular, referee #1 finds that the proposed effect of espin on confined cell migration through filopodia is currently not sufficiently supported by the data, which is a critical concern to address for further consideration at this journal.

Should you be able to address the referee concerns satisfactorily, we would like to invite you to submit a revised manuscript. Please revise your manuscript with the understanding that the referee concerns (as in their reports) must be fully addressed and their suggestions taken on board. Please address all referee concerns in a complete point-by-point response. Acceptance of the manuscript will depend on a positive outcome of a second round of review. It is EMBO reports policy to allow a single round of major experimental revision only and acceptance or rejection of the manuscript will therefore depend on the completeness of your responses included in the next, final version of the manuscript.

We realize that it is difficult to revise to a specific deadline. In the interest of protecting the conceptual advance provided by the work, we recommend a revision within 3 months. Please discuss the revision progress ahead of this time with me if you require more time to complete the revisions, or if you have questions or comments regarding the revision (also by video chat).

1. A data availability section providing access to data deposited in public databases is missing (where applicable).
2. Your manuscript contains statistics and error bars based on $n=2$. Please use scatter plots in these cases.

You can submit the revision either as a Scientific Report or as a Research Article. For Scientific Reports, the revised manuscript can contain up to 5 main figures and 5 Expanded View figures, and it should not exceed 27000 characters. If the revision leads to a manuscript with more than 5 main figures it will be published as a Research Article. In this case the Results and Discussion section should be separate. If a Scientific Report is submitted, these sections have to be combined. This will help to shorten the manuscript text by eliminating some redundancy that is inevitable when discussing the same experiments twice. In either case, all materials and methods should be included in the main manuscript file.

4) a .docx formatted letter INCLUDING the reviewers' reports and your detailed point-by-point responses to their comments. As part of the EMBO publication's Transparent Editorial Process, EMBO reports publishes online a Review Process File (RPF) to accompany accepted manuscripts. This File will be published in conjunction with your paper and will include the referee reports, your point-by-point response and all pertinent correspondence relating to the manuscript.

<https://www.embopress.org/page/journal/14693178/authorguide#transparentprocess>

5) a complete author checklist, which you can download from our author guidelines <https://www.embopress.org/page/journal/14693178/authorguide>. Please insert information in the checklist that is also reflected in the manuscript. The completed author checklist will also be part of the RPF.

6) Please note that all corresponding authors are required to supply an ORCID ID for their name upon submission of a revised manuscript (<<https://orcid.org/>>). Please find instructions on how to link your ORCID ID to your account in our manuscript tracking system in our Author guidelines <<https://www.embopress.org/page/journal/14693178/authorguide#authorshipguidelines>>

Additional information on source data and instruction on how to label the files are available: <https://www.embopress.org/page/journal/14693178/authorguide#sourcedata>

9) Our journal encourages inclusion of *data citations in the reference list* to directly cite datasets that were re-used and obtained from public databases. Data citations in the article text are distinct from normal bibliographical citations and should directly link to the database records from which the data can be accessed. In the main text, data citations are formatted as follows: "Data ref: Smith et al, 2001" or "Data ref: NCBI Sequence Read Archive PRJNA342805, 2017". In the Reference list, data citations must be labeled with "[DATASET]". A data reference must provide the database name, accession number/identifiers and a resolvable link to the landing page from which the data can be accessed at the end of the reference. Further instructions are available at <http://www.embopress.org/page/journal/14693178/authorguide#referencesformat>

- the name of the statistical test used to generate error bars and P values,
- the number (n) of independent experiments (please specify technical or biological replicates) underlying each data point,
- the nature of the bars and error bars (s.d., s.e.m.),
- If the data are obtained from n Program fragment delivered error `Can't locate object method "less" via package "than" (perhaps you forgot to load "than"?) at //ejpvfs23/sites23b/embor_www/letters/embor_decision_revise_and_review.txt line 56.' 2, use scatter blots showing the individual data points.

12) Please also note our reference format:

13) All Materials and Methods need to be described in the main text using our 'Structured Methods' format, which is required for all research articles. According to this format, the Methods section includes a Reagents and Tools Table (listing key reagents, experimental models, software and relevant equipment and including their sources and relevant identifiers) followed by a Methods and Protocols section describing the methods using a step-by-step protocol format. The aim is to facilitate adoption of the methodologies across labs. More information on how to adhere to this format as well as a downloadable template (.docx) for the Reagents and Tools Table can be found in our author guidelines:

I look forward to seeing a revised version of your manuscript when it is ready. Please let me know if you have questions or comments regarding the revision.

Kind regards,

Deniz

Deniz Senyilmaz Tiebe, PhD
Senior Scientific Editor
EMBO Reports

Referee #1:

In the study entitled "Espin enhances confined cell migration by promoting filopodia formation and 1 contributes to cancer metastasis", Yan Wang and colleagues identify espin as an active regulator for confined cell migration and show that overexpression of espin is associated with the formation of metastases.

Several major critical points can be identified by reading the manuscript:

An important issue is the quantitative analysis shown in several Figures: a major concern is about the quality of the data used for several quantifications presented in the manuscript. Often the exact methods for quantification can not be found in the text/Methods. Specific examples (not exhaustive) are presented.

As one example, in Figure 4E-F, how were filopodia analyzed to build the graphs shown in these panels? The quality of images to measure filopodial length (examples in panel 4E) makes it hard not to miss important information/filopodia that may be lost/undetectable in this low resolution images.

Another example is Figure 4L+M: it is hard to imagine that the very subtle (significant) difference shown in panel M can be obtained from the poor quality images shown in the Figure (2 μ M condition). Precise filament length is very hard (if not impossible) to measure from these images.

In Figure 5E/F, quantification from the same experiments of wt espin should be included together with control and espin mutant.

Is the difference shown in Figure 5H significant?

In panel 5K, from the images shown, partial colocalization of espin and paxillin (used here as a focal adhesion marker) can not be excluded (it is indeed detectable). Also, how can the authors be sure that the espin spots are filopodia with this type of resolution? Espin is not limited to protrusions in cells: see for example Figure 4A, where espin is copiously detected also at the cell cortex. Therefore, the conclusion that filopodia mediate movement in confined environments is far from being demonstrated (but also hard to be suggested) by the data presented.

In Figure 5L: staining of espin is very different from examples shown in previous images: why is it so? Moreover, tension marker appears to be predominantly associated to sites different from filopodia (look rather cortical sites) in this image: given, as already said, the widespread distribution of espin (not limited to filopodia), it is hard to support the conclusion that "espin overexpressing cells formed excessive filopodia in front of the nucleus towards the cell front and along the sides, for cells to pull on and paddle

with".

In general, the data supporting the hypothesis that "Espn enhances confined cell migration by promoting filopodia formation" (in the title and at other points of the manuscript) is poorly supported by the data presented in the manuscript.

Other important points:

Often it is left to the reader's imagination to extrapolate information from the very succinct Figure legends, that should in many cases be revised and made self-explanatory with respect to the data presented in the respective Figures. Several examples that need revision for proper description can be found in this manuscript.

Nowhere in the text can be found a clear explanation of the use of the drugs CK666 (Arp2/3 Complex inhibitor) and SMIFH2 (formin inhibitor).

Some grammar checking is required.

Referee #2:

The manuscript "Espn enhances confined cell migration by promoting filopodia formation and contributes to cancer metastasis" by Wang, Shi, Liu, Hu, Wang, Chen, Yang, Wai, Chen, Liang, Liu, Liu, and Wu used engineering tools to study confined migration and specifically, based on results of a screen, the role of the actin bundling protein espin in this process. This is a highly interesting work that will find a good level of readership in EMBO reports.

Minor concerns:

The transwell assays are indeed a good tool for mimicking the confining microenvironment in the body, and knowing their cross-sectional area is important. However, the lengths of these constrictions are not mentioned.

In Figure 1, you quantify the velocity of cells that enter the microchannels. What about the number of cells that entered? Based on the transwell assay, I would expect more cells to enter the microchannels in the espin OE cells. Is this what you observed?

In the survival data in Figure 2B, it only looks like high espin cells are associated with lower survival at intermediate time points. This can be clarified in the text.

It is unclear to me how the images in Fig 2G are connected to the foci measurement in Fig 2H.

In Figure 5, the authors make the point that the cavity formation could be a function of increased speed at the leading edge, and contrast that with the nucleus. However, it is unclear where the nucleus is in these images.

In figure 5L, the Espn stain/reporter has a very distinct pattern that is not seen in previous Espn images (5J and 5G, for example). Does the DNA sensor surface induce this phenotype?

The grammar is poor throughout most of the manuscript, and thus it would benefit from editing to ensure that meanings are conveyed accurately.

Referee #1:

In the study entitled "Espin enhances confined cell migration by promoting filopodia formation and contributes to cancer metastasis", Yan Wang and colleagues identify espin as an active regulator for confined cell migration and show that overexpression of espin is associated with the formation of metastases.

We thank the reviewer for the valuable feedback to improve the quality of our manuscript. We have taken the reviewer's suggestions and improved our figures and quantification methods. As for the "Espin enhances confined cell migration by promoting filopodia formation", we have carried out experiments to strengthen the conclusion. The detailed results were attached and discussed below. We feel that this new version of manuscript is much improved and our conclusions are more consolidated this way.

Several major critical points can be identified by reading the manuscript:

An important issue is the quantitative analysis shown in several Figures: a major concern is about the quality of the data used for several quantifications presented in the manuscript.

We apologize for the low resolution of figures in our previous manuscript due to unintended figure compression during format conversion. In the revised manuscript, we have updated all relevant figures to ensure the details in the figures are discernible, including but not limited to filopodia and actin filaments. Most of the figures in our manuscript are with the pixel size of 103.2 nm. Revised figures have been attached below in contrast with former figures.

Often the exact methods for quantification can not be found in the text/Methods. Specific examples (not exhaustive) are presented.

We apologize for not providing detailed quantification methods in our previous manuscript. We have added related descriptions in the Materials & Methods section in our revised manuscript (line 441-449), including the quantification of filopodial length and F-actin width.

As one example, in Figure 4E+F, how were filopodia analyzed to build the graphs shown in these panels? The quality of images to measure filopodial length (examples in panel 4E) makes it hard not to miss important information/filopodia that may be lost/undetectable in this low resolution images.

We apologize again for submitting the poor-quality images after figure compression and we have updated the figures with high-resolution ones. In comparison with previous Figure 4E (Fig. R1.1A), filopodia are discernible in the revised Figure 4E (Fig. R1.1B). To quantify filopodial number, we included protrusions labelled by F-actin. Then, unpaired t-test was used to assess the statistical significance between control and espin OE group. The statistical result of filopodial number were shown in Figure 4F.

To measure filopodial length, the filopodia were detected from espin or fascin channel such as Figure 4I (Fig. R1.1D). We agree with the reviewer that detailed information could not be detectable from the low-resolution images (Fig. R1.1C). We have updated with high-resolution images (Fig. R1.1D). To quantify filopodial length, we first measured the fluorescence intensity of background (box 1) and cytoplasm (box 3) as references (Fig. R1.1E). Then, intensity along the filopodia (the red triangle marked) were obtained (box 2). The filopodia origins from where the intensity was above the cytoplasm ($x=1.548 \mu\text{m}$) and reaches where the intensity was above the background ($x'=5.0568 \mu\text{m}$). The filopodial length was calculated as the distance between x and x' . The above quantification process was diagrammed as below (Fig. R1.1E).

Figure R1.1

Figure R1.1. Revised images for filopodial number and length quantification

A & B: Figure 4E before and after revision. Representative images showing F-actin of control and espin OE cells, triangles mark representative filopodia. Cell areas in white boxes are enlarged. Scale bar: 20 μm .

C & D: Figure 4I before and after revision. Representative images showing signals of espin or fascin in corresponding OE cells. "Bottom" shows the bottom layer while "apical" shows the maximum projection of dorsal layers. Cell areas in white boxes are enlarged. Scale bar: 20 μm .

E: Quantification steps of filopodial length.

Another example is Figure 4L+M: it is hard to imagine that the very subtle (significant) difference shown in panel M can be obtained from the poor quality images shown in the Figure (2 μM condition). Precise filament length is very hard (if not impossible) to measure from these images.

We thank the reviewer for pointing this out and we fully agree that precise width quantification of actin filaments could not be obtained from poor-quality figures. Here, we show respective high-resolution images of Figure 4L (Fig.

R1.2B) which have replaced previous images (Fig. R1.2A) in our revised manuscript. In the revised images, actin filaments are recognizable. We could conclude that actin filaments with purified espin added were thicker than that with fascin.

As for the width quantification of F-actin in Figure 4L, we first measured the fluorescence intensity of the background (solid line 1) as a reference. We then drew lines perpendicular to the direction of F-actin (solid line 2) and measured intensity along the lines. The widths of F-actin were defined as the length with intensity above the background. The above quantification process was diagrammed as below (Fig. R1.2C). The statistical analysis, using an unpaired t-test, showed a p-value of less than 0.0001, indicating a highly significant difference between the groups. The mean width of F-actin with espin or fascin added were 0.6229 μm and 0.5301 μm , respectively (Fig. R1.2D).

Figure R1.2

Figure R1.2. Revised images for F-actin width quantification

A & B: Figure 4L before and after revision. Representative images showing actin filaments with purified espin or fascin added. Cell areas in white boxes are enlarged.

C: Detailed steps of F-actin width quantification.

D: Result of t-test between 2 μM espin and fascin groups after measuring F-actin widths in (B).

In Figure 5E/F, quantification from the same experiments of wt espin should be included together with control and espin mutant.

We thank the reviewer for the suggestion and we have included wt espin group, both in filopodial number quantification and confined migration ability assessment. After number quantification, one-way ANOVA was applied to determine that it was statistically significant among the three groups. Then, unpaired t-test was used to compare individual two groups. In contrast to the control group, wt espin OE promoted filopodia formation (Fig. R1.3A,B) and enhanced confined migration (Fig. R1.3C,D). These two phenotypes were abolished by espin ABM depletion

(Fig. R1.3A-D), suggesting that the active role of espin in filopodia formation and confined migration was dependent of its ABM domain. These figures have been incorporated into the revised manuscript (Fig. 5E-H).

Figure R1.3

Figure R1.3. Espin promotes filopodia formation and contributes to confined migration dependent of ABM domain

A: Representative images showing F-actin of cells stably transfected with vehicle, wt espin or espin Δ ABM. Cell areas in white boxes are enlarged. Triangles mark representative filopodia. Scale bar: 20 μ m in upper images and 5 μ m in bottom enlarged images, respectively.

B: Quantification of filopodial number per cell in (A). Data are shown as mean \pm SD. n(Ctrl) = 16, n(Espin OE) = 17, n(Espin Δ ABM) = 16. Significance was tested using one-way ANOVA and unpaired Student's t-test.

C: Representative images of bottom cells on 5 μ m transwell, cells were stably transfected with vehicle, wt espin or espin Δ ABM. The black puncta represent nuclei stained with Hoechst. Scale bar: 50 μ m.

D: Cell number quantification in (C). Data are shown as mean \pm SD. n(Ctrl) = 7, n(Espin OE) = 7, n(Espin Δ ABM) = 7. Significance was tested using one-way ANOVA and unpaired Student's t-test.

Is the difference shown in Figure 5H significant?

We apologize for showing the results from just one experiment. Here, we showed representative images from another independent experiment (Fig. R1.4A), with the white arrow marked cavity between cell body and microchannels. Moreover, we calculated the cell ratio with cavity from three independent experiments (Fig. R1.4B) and applied unpaired t-test. The p value was 0.0182, suggesting that the difference was statistically significant. The quantification result has been incorporated into our revised manuscript (Fig. 6B).

Figure R1.4

Figure R1.4. Espin OE cells form cavity with microchannels more than control cells

A: Representative images of control and espn OE cells in confined microchannels. Image left shows control cell labeled by LifeAct while right shows espn-EGFP cell. Scale bar: 20 μm .

B: Ratio quantification of cells forming cavities with the microchannels. Data are shown as mean \pm SD. $N(\text{Ctrl}) = 3$, $N(\text{Espin OE}) = 3$. Significance was tested using unpaired Student's t-test.

In panel 5K, from the images shown, partial colocalization of espn and paxillin (used here as a focal adhesion marker) can not be excluded (it is indeed detectable). Also, how can the authors be sure that the espn spots are filopodia with this type of resolution?

We apologize for the poor quality of images in previous manuscript, resulting from figure compression and low fluorescence intensity, making the specific signals unrecognizable. In the revised images, the espn filaments and paxillin plaques were recognizable (Fig. R1.5A). We observed that most of the espn filaments were not colocalized with paxillin, only except for where the white triangles marked. These images have been included in the revised manuscript (Fig. 6F). To better describe the localization correlation of espn and paxillin in confined cells, we calculated the Pearson correlation coefficients of espn and paxillin (Fig. R1.5B). To filter unspecific background signals, a manual intensity threshold was used when using the Coloc2 in ImageJ to calculate Pearson coefficients. The data were in accordance with normal distribution as shown by the p value of Anderson-Darling test (Fig. R1.5C). The mean value was approximately to zero, suggesting that there was no correlation between espn and paxillin localization (Fig. R1.5D).

Figure R1.5

Figure R1.5. No localization correlation between espion and paxillin in confined cells

A: Representative images of paxillin signals in espion OE cells in confined channels. Cell areas in white boxes are enlarged. Scale bar: 20 μm .

B: The Pearson coefficient of espion and paxillin in confined cells.

C: The result of Anderson-Darling test showing that the data in (B) are in accordance with normal distribution.

D: The descriptive statistics of data in (B).

To make sure that espion localized on filopodia in confined cells, we also transfected filopodial marker fascin or myosin-X. The results showed colocalization of espion and fascin on filopodia (Fig. R1.6A). Besides, myosin-X localized on the tip of espion protrusions, suggesting that espion localized on filopodia (Fig. R1.6B). These figures have been incorporated into the revised manuscript (Fig. 6D, Fig. EV4B).

Figure R1.6

Figure R1.6. Espin localizes on filopodia in confined cells

A: Representative images of fascin colocalization with espn in confined channels. Espn OE cells were transfected with fascin. Cell areas in white boxes are enlarged. Triangles mark representative filopodia with espn and fascin colocalized. Scale bar: 20 μm .

B: Representative images of myosin-X localization and espn in confined channels. Espn OE cells were transfected with myosin-X. Cell areas in white boxes are enlarged. Triangles mark representative myosin-X at the tip of espn-protrusions. Scale bar: 20 μm .

Espin is not limited to protrusions in cells: see for example Figure 4A, where espn is copiously detected also at the cell cortex. Therefore, the conclusion that filopodia mediate movement in confined environments is far from being demonstrated (but also hard to be suggested) by the data presented.

We thank the reviewer for the helpful reminding to improve our manuscript. Following the reviewer's concern, we have calculated the cell ratio with cortex localization of espn from four independent experiments and found that lower than 5% cells showed partial cortex localization of espn (Fig. R1.7A). We apologize for providing such an unrepresentative image in previous Figure 4A. Here, we provided some representative images of espn localization (Fig. R1.7B) and we have updated the Figure 4A in the revised manuscript (Fig. R1.7C). In the revised images, espn exhibited colocalization with F-actin along filopodia as the triangles pointed out, while was absent from cell cortex where the arrows marked. In the last submission, such images without cortex localization of espn could also be seen, such as Figure 4G and 4I, which were also attached below (Fig. R1.7D). Besides, the localization of espn in other journals were also concentrated at cell protrusions (Fig. R1.7E) [1, 2]. In the bottom image of Fig. R1.6E, espn localization was absent from cell cortex labelled by F-actin in red.

Moreover, we wondered whether the slight localization of espn on cortex could have an effect. Cell cortex was stained using phalloidin and the intensities were quantified. One-way ANOVA was applied to determine that it was not statistically significant among the five groups, indicating that altered expression level of espn did not influence

cortex content (Fig. R1.7F,G). Thus, we concluded that enhanced confined migration by espin was independent of cell cortex although espin showed cortex localization in minority cells. We feel that our conclusion that “Espin enhances confined cell migration by promoting filopodia formation” is strengthened this way.

Figure R1.7

Figure R1.7. Espin does not show significant cortex localization and does not interfere with cell cortex

A: Cell proportion with cortex localization of espin.

B: Representative images showing espin localization on filopodia. Scale bar: 20 μm .

C: Representative images showing colocalization of espin and F-actin on filopodia rather than cortex. Triangles and arrows mark representative filopodia and cell cortex, respectively. Scale bar: 5 μm .

D: Figure 4G and 4I in previous manuscript also show espin localization on filopodia. Cell areas in white boxes are enlarged. Scale bar: 20 μm .

E: Representative images showing espin localization on filopodia rather than cortex in previous reports. Cell areas in white boxes are enlarged. Scale bar: 5 μm .

F: Representative images of cell cortex. Scale bar: 20 μm .

G: Fluorescence intensity quantification of cell cortex in (F). Data are shown as mean \pm SD. n(Ctrl) = 15, n(Espin OE) = 19, n(shCtrl) = 22, n(shEspin-1) = 10, n(shEspin-2) = 10. Significance was tested using one-way ANOVA.

In Figure 5L: staining of espin is very different from examples shown in previous images: why is it so?

We thank the reviewer for pointing this out. After careful measurement, we found that the bright areas in Figure 5L were overexposed as shown by the consistent intensity (Fig. R1.8A), which made the filaments indistinguishable. We have improved our figures in the revised manuscript (Fig. 6G). As shown below, the tension signals exhibited partial colocalization with espin-positive filopodia at the leading edge and displayed a sparse dot pattern at the tip of side filopodia (Fig. R1.8B).

Figure R1.8

Figure R1.8. Tension signals along with or at the tip of espin-positive filopodia

A: Previous Figure 5L. The bottom image shows espin intensity along the solid line in the upper image. The area between the black arrows is overexposed as shown by the consistent intensity. Scale bar: 5 μm .

B: Representative images of espin and 17pN DNA-based tension probe signals in confined microchannels. Images above show the tension signals along filopodia at the leading edge while the bottom show signals at the tip of side filopodia, as the triangles mark. Scale bar: 5 μm .

Moreover, tension marker appears to be predominantly associated to sites different from filopodia (look rather cortical sites) in this image: given, as already said, the widespread distribution of espin (not limited to filopodia), it

is hard to support the conclusion that "espin overexpressing cells formed excessive filopodia in front of the nucleus towards the cell front and along the sides, for cells to pull on and paddle with".

We apologize for the poor quality of images in last submitted manuscript. The side filopodia were concentrated (frequency about 0.82 μm) and short (mean length at 0.63 μm). Unfortunately, the low-resolution images blurred the morphology of side filopodia, which were originally discontinuous along the cell periphery as clarified by espin intensity. The tension marker localized at the tip of side filopodia marked by the white arrowheads (Fig. R1.9A). Besides, the colocalization of espin and fascin on side filopodia were shown, indicating espin's filopodial location (Fig. R1.9B).

Figure R1.9

Figure R1.9. Espin localizes on filopodia rather than cortex in confined cells

A: Previous Figure 5L. The right image shows espin intensity along the white arrow in the enlarged area. The white arrowheads and black arrows mark corresponding filopodia with tension markers at the tip. Scale bar: 5 μm .

B: Representative images of fascin and espin localization in confined channels. Espin OE cells were transfected with fascin. The right panel shows fascin and espin intensity along the white arrow. The white arrowheads and black arrows mark corresponding filopodia with fascin and espin colocalized. Scale bar: 20 μm .

In general, the data supporting the hypothesis that "Espin enhances confined cell migration by promoting filopodia formation" (in the title and at other points of the manuscript) is poorly supported by the data presented in the manuscript.

We thank the reviewer for noticing the cortex localization of espin and pointing this out to make our conclusions more accurate. In our previous submission, we proposed the hypothesis that “Espin enhances confined cell migration by promoting filopodia formation”, which was suggested by the impaired confined migration upon filopodial number decreasing, either by fascin or myosin-X knockdown. Following the reviewer’s suggestions, we also assessed the localization and effect of espin on cortex. We found that espin did not show significant cortex localization or interfere with cell cortex. In confined cells, we also observed that espin localized at filopodia labelled by filopodial markers fascin and myosin-X. In sum, espin localized mainly on filopodia and induced abundant filopodia formation, thus enhancing confined migration.

Other important points:

Often it is left to the reader’s imagination to extrapolate information from the very succinct Figure legends, that should in many cases be revised and made self-explanatory with respect to the data presented in the respective Figures. Several examples that need revision for proper description can be found in this manuscript.

We apologize for not providing detailed explanations in Figure legends and we have revised the text and figure legends throughout the manuscript so that the conclusions could be suggested accurately.

Nowhere in the text can be found a clear explanation of the use of the drugs CK666 (Arp2/3 Complex inhibitor) and SMIFH2 (formin inhibitor).

We thank the reviewer for pointing this out. Since espin OE induced abundant filopodia formation, we wondered the underlying mechanisms. In the filopodia formation, the convergent elongation model based on Arp2/3 complex and the de novo filament nucleation model dependent of formins have been proposed [3-5]. To investigate which model espin-enhanced filopodia formation was dependent of, we used Arp2/3 and formin inhibitors and quantified filopodia number after drug treatment. And we have added explanations of the use of Arp2/3 and formin inhibitors in our revised manuscript (line 159-166).

Some grammar checking is required.

We appreciate the reviewer for careful reading. We have checked the grammar throughout the manuscript and polished the entire text with changes highlighted (example in line 77, 147 and 178). These changes will not influence the content and framework of the paper. We sincerely appreciate the reviewer’s efforts in helping us improve our work.

References

1. Chou, S.W., et al., *Fascin 2b is a component of stereocilia that lengthens actin-based protrusions*. PLoS One, 2011. **6**(4): p. e14807.
2. Loomis, P.A., et al., *Espin cross-links cause the elongation of microvillus-type parallel actin bundles in vivo*. J Cell Biol, 2003. **163**(5): p. 1045-55.
3. Evangelista, M., et al., *Formins direct Arp2/3-independent actin filament assembly to polarize cell growth in yeast*. Nature Cell Biology, 2001. **4**(1): p. 32-41.
4. Svitkina, T.M., et al., *Mechanism of filopodia initiation by reorganization of a dendritic network*. The Journal of Cell Biology, 2003. **160**(3): p. 409-421.
5. Mattila, P.K. and P. Lappalainen, *Filopodia: molecular architecture and cellular functions*. Nat Rev Mol Cell Biol, 2008. **9**(6): p. 446-54.

Referee #2:

The manuscript "Espin enhances confined cell migration by promoting filopodia formation and contributes to cancer metastasis" by Wang, Shi, Liu, Hu, Wang, Chen, Yang, Wai, Chen, Liang, Liu, Liu, and Wu used engineering tools to study confined migration and specifically, based on results of a screen, the role of the actin bundling protein espin in this process. This is a highly interesting work that will find a good level of readership in EMBO reports.

We thank the reviewer for the recognition of our work. We also appreciate the reviewer for the insightful comments and suggestions. Following the suggestions, we carried out relevant experiments to improve our manuscript and supplemented detailed description of the experimental materials in the revised manuscript. The detailed revisions were attached and discussed below.

Minor concerns:

The transwell assays are indeed a good tool for mimicking the confining microenvironment in the body, and knowing their cross-sectional area is important. However, the lengths of these constrictions are not mentioned.

We apologize for not providing detailed size information about the constrictions in the transwell in previous manuscript. In the revised manuscript, we have added the lengths of constrictions (line 346 and line 589). According to the Corning website, the thickness of transwell membrane is 10 μm , which corresponds to the length of constrictions. Additionally, we also measured the length of constrictions using confocal z-stacks taken with 1 μm steps. Based on the steps between upper and bottom cells, the measured length of constrictions is about 11 μm (Fig. R2.1A), which is a comparable size as described theoretically.

Figure R2.1

A

Figure R2.1. Length of constrictions in transwell

A: Confocal z-stacks taken with 1 μm steps. Stack 1 shows bottom nuclei on transwell while stack 12 shows upper nuclei. Dashed circles mark representative bottom and upper nuclei. Scale bar: 20 μm .

In Figure 1, you quantify the velocity of cells that enter the microchannels. What about the number of cells that entered? Based on the transwell assay, I would expect more cells to enter the microchannels in the espin OE cells. Is this what you observed?

We sincerely appreciate the reviewer's suggestions. Since the constriction length of transwell was comparable as cell size, we agree with the reviewer that espin OE cells are expected to enter the microchannels more than control cells. Before seeding cells, we punched the microchannels to produce sample holes. To avoid the manual variability between individual microchannels, we mixed an equal number of control and espin OE cells after staining nuclei with Hoechst. The cells were seeded together. Indeed, we observed that most of the entered cells were espin OE

cells (with cyan cell body and blue nucleus, as the arrow marked) rather than control cells (just with blue nucleus, as the triangle marked) (Fig. R2.2A). We calculated the percentage of control or espin OE cells in the entered group and found that more espin OE cells entered the microchannels (Fig. R2.2B). We have incorporated these updated images into our revised manuscript (Fig. 1I, Fig. EV1D).

Figure R2.2

Figure R2.2. Espin OE cells enter microchannels more than control cells

A: Representative images showing control (just blue) and espin OE cells (cyan and blue) in microchannels. Scale bar: 20 μ m.

B: Quantification of cell percentage in (A). Data are shown as mean \pm SD. N(Ctrl) = 5, N(Espin OE) = 5. Significance was tested using unpaired Student's t-test.

In the survival data in Figure 2B, it only looks like high espin cells are associated with lower survival at intermediate time points. This can be clarified in the text.

We thank the reviewer for this insightful suggestion. The patients with higher espin levels exhibited lower survival within the first 150 months but showed similar survival rates at intermediate time points, suggesting that espin could interfere with cancer aggressiveness at the early period. We have included this conclusion in the revised manuscript (line 118-119).

It is unclear to me how the images in Fig 2G are connected to the foci measurement in Fig 2H.

We apologize for not providing a detailed explanation of the metastatic foci quantification in last manuscript. As shown in the enlarged image (Fig. R2.3A), black puncta are visible on the lung surface, such as those marked by dashed circles. These puncta were counted as metastatic foci. We have added the description of metastatic foci in our revised manuscript (line 125-126).

Figure R2.3

A

Figure R2.3. Metastatic foci quantification

A: Metastatic foci are morphologically circular and black to be visible on the lung surface, as shown in dashed circles.

In Figure 5, the authors make the point that the cavity formation could be a function of increased speed at the leading edge, and contrast that with the nucleus. However, it is unclear where the nucleus is in these images.

We apologize for not providing the nuclear position in previous Figure 5G. In the revised images, we presented and highlighted the nucleus with dashed lines (Fig. R2.4A). These updated images have been included in our revised manuscript (Fig. 6A).

Figure R2.4

A

Figure R2.4. Espin OE cells form cavity with microchannels at the cell leading part

A: Representative images of control and espin OE cells in confined microchannels with 5 μm width. Images left show control cell labeled by LifeAct while right images show espin-EGFP cell. The nuclei were presented alone and highlighted with dashed lines in merge images. Arrows mark the cavity between cell body and the channel. Scale bar: 20 μm .

In figure 5L, the Espin stain/reporter has a very distinct pattern that is not seen in previous Espin images (5J and 5G, for example). Does the DNA sensor surface induce this phenotype?

We thank the reviewer for pointing this out. After careful measurement, we found that the bright areas in Figure 5L were overexposed as shown by the consistent intensity (Fig. R2.5A), which made the filaments indistinguishable. We have improved our figures in the revised manuscript (Fig. 6G). As shown below, the tension signals exhibited partial colocalization with espin-rich filopodia at the leading edge and displayed a sparse dot pattern at the tip of side filopodia (Fig. R2.5B). Besides, the espin OE cells in confined microchannels with DNA tension sensor surface also showed cavities with microchannels and espin-positive filopodia, similar to that without DNA tension sensor coated (Fig. R2.5C). Thus, the DNA tension sensor did not affect the morphology or espin localization of confined cells.

Figure R2.5

Figure R2.5. Tension signals along or at the tip of filopodia without affecting espin pattern

A: Previous Figure 5L. The bottom image shows espin intensity along the solid line in the upper image. The area between the black arrows is overexposed as shown by the consistent intensity. Scale bar: 5 μm .

B: Representative images of espin and 17pN DNA-based tension probe signals in confined microchannels. Images above show the tension signals along filopodia at the leading edge while the bottom show signals at the tip of side filopodia, as the triangles mark. Scale bar: 5 μm .

C: Representative images of espin OE cells in confined microchannels with or without 17pN DNA-based tension probe coated. Scale bar: 20 μm .

The grammar is poor throughout most of the manuscript, and thus it would benefit from editing to ensure that meanings are conveyed accurately.

We thank the reviewer for this suggestion. During revision, we have carefully checked and corrected grammatical errors throughout the manuscript and polished the entire text with changes highlighted (example in line 77, 147 and 178). We feel that this new version of manuscript is much improved. Again, we sincerely appreciate the reviewer's efforts in helping us improve our work.

Dear Congying,

Thank you for submitting your revised manuscript. It has now been seen by both of the original referees. I apologize for this unusual delay in getting back to you. As mentioned, it took longer than anticipated to receive the referee reports.

As you can see, both referees find that the study is significantly improved during revision and recommend publication. However, I need you to address the points below before I can accept the manuscript.

- Please consider moving the Figures R2.1 and R2.3 to the EV Figures as per referee #2. Also, please address the remaining minor concern of referee #2.
- We note that the ORCID iD of Dr. Peng Shi is not provided to the manuscript submission system. EMBO Press policy asks for all corresponding authors to link to their ORCID iDs. Please see the Guide to Authors here: <https://www.embopress.org/page/journal/14693178/authorguide#authorshipguidelines>

In order to link your ORCID iD to your account in our manuscript tracking system, please do the following:

1. Click the 'Modify Profile' link at the bottom of your homepage in our system.
2. On the next page you will see a box halfway down the page titled ORCID*. Below this box is red text reading 'To Register/Link to ORCID, click here'. Please follow that link: you will be taken to ORCID where you can log in to your account (or create an account if you don't have one)
3. You will then be asked to authorise Wiley to access your ORCID information. Once you have approved the linking, you will be brought back to our manuscript system.

We regret that we cannot do this linking on your behalf for security reasons.

- As per our format requirements, in the reference list, citations should be listed in alphabetical order and then chronologically, with the authors' surnames and initials inverted; where there are more than 10 authors on a paper, 10 will be listed, followed by 'et al.'. Please see <https://www.embopress.org/page/journal/14693178/authorguide#referencesformat>
- Please remove the 'Author Contributions' section from the manuscript text.
- Please remove the synopsis image from the manuscript text and submit it as a separate jpg or tif file with the correct dimensions (i.e. 550 (width) x 300-600 (height) pixels).
- Please make the dataset GSE286976 publicly available and remove the reviewer token from the manuscript.
- In the Data Availability section, please provide a link that directly resolves to the GSE286976 dataset.
- Our production/data editors have asked you to clarify several points in the figure legends:
 - o Figure Legends (main + EV): 1. Please define the annotated p values ****/**/*/* as well as provide the exact p-values for the same in the legend of figure 2A as appropriate.
 - o Please note that the exact p values are not provided in the legends of figures 1E, G, I; 4F, H, M; 5D, F, H.
 - o Please indicate the statistical test used for data analysis in the legends of figures 1B, 2A, B.
 - o Please note that the box plots need to be defined in terms of minima, maxima, centre, bounds of box and whiskers, and percentile in the legends of figures 2A.
 - o Please note that information related to n is missing in the legends of figures 2A.
 - o Please note that the error bars are not defined in the legends of figures EV1 B, C.
 - o Please note that the scale bar needs to be defined for figure 3E.
- Papers published in EMBO Reports include a 'synopsis' and 'bullet points' to further enhance discoverability. Both are displayed on the html version of the paper and are freely accessible to all readers. The synopsis includes a short standfirst summarizing the study in 1 or 2 sentences (max 35 words) that summarize the paper and are provided by the authors and streamlined by the handling editor. I would therefore ask you to include your synopsis blurb and 3-5 bullet points listing the key experimental findings.

Thank you again for giving us to consider your manuscript for EMBO Reports, I look forward to your minor revision.

Kind regards,

Deniz

--

Deniz Senyilmaz Tiebe, PhD
Senior Scientific Editor
EMBO Reports

Referee #1:

After careful revision, the study "Espin enhances confined cell migration by promoting filopodia formation and contributes to cancer metastasis" has been significantly improved by Yan Wang and colleagues, with images of adequate quality to support quantification and main conclusions. I believe that in its present form the manuscript is adequate for publication in EMBO Reports.

Referee #2:

Overall the manuscript has improved significantly. My only final suggestions are to:

- 1) In general I am against the use of 'reviewer-only' figures. As reviewers, our points of clarification represent where the scientific community might also need points of clarification. Figure R2.1 and R2.3 are not in the final version of the manuscript, even though there is plenty of space in the supplemental figures and (in the case of R2.3) in Figure 2 itself (the current images of whole lungs are not particularly helpful for communicating findings).
- 2) The new figure 6G is missing the 'leading edge' and 'side' labels on the left side of the images.

All editorial and formatting issues were resolved by the authors.

Prof. Congying Wu
Peking University Health Science Center
Institute of Systems Biomedicine ,School of Basic Medical Sciences
38th,Xueyuan road, Haidian district
Beijing, Beijing 100191
China

Dear Congying,

Thank you for submitting your revised manuscript. I have now looked at everything and all is fine. Therefore, I am very pleased to accept your manuscript for publication in EMBO Reports.

Congratulations on a nice work!

Kind regards,

Deniz

--

Deniz Senyilmaz Tiebe, PhD
Senior Scientific Editor
EMBO Reports

--
